# VLP: Vision-Language Preference Learning for Embodied Manipulation

## Abstract

Reward engineering is one of the key challenges in Reinforcement Learning (RL). Preference-based RL effectively addresses this issue by learning from human feedback. However, it is both time-consuming and expensive to collect human preference labels. In this paper, we propose a novel **V**ision-**L**anguage **P**reference learning framework, named **VLP**, which learns a vision-language preference model to provide preference feedback for embodied manipulation tasks. To achieve this, we define three types of language-conditioned preferences and construct a vision-language preference dataset, which contains versatile implicit preference orders without human annotations. The preference model learns to extract language-related features, and then serves as a preference annotator in various downstream tasks. The policy can be learned according to the annotated preferences via reward learning or direct policy optimization. Extensive empirical results on simulated embodied manipulation tasks demonstrate that our method provides accurate preferences and generalizes to unseen tasks and unseen language instructions, outperforming the baselines by a large margin. The code and videos of our method are available on the website: `https://VLPref.github.io`.

## 1 Introduction

Reinforcement Learning (RL) has made great achievements recent years, including board games (Silver et al., 2017; 2018), autonomous driving (Kiran et al., 2021; Zhou et al., 2021), and robotic manipulation (Kober et al., 2013; Andrychowicz et al., 2020; Chen et al., 2022). However, one of the key challenges to apply RL algorithms is reward engineering. First, designing an accurate reward function requires large amount of expert knowledge. Second, the agent might hack the designed reward function (Hadfield-Menell et al., 2017), obtaining high returns without completing the task. Also, it is difficult to obtain reward functions for subjective human objectives.

To address the above issues, a variety of works leverage expert demonstrations for imitation learning (IL) (Ho & Ermon, 2016; Torabi et al., 2018). Nevertheless, expert demonstrations are often expensive and the performance of IL is limited by the quality of the demonstrations. Another line of work leverages Vision-Language Models (VLMs) to provide multi-modal rewards for downstream policy learning (Nair et al., 2023; Ma et al., 2023a; Rocamonde et al., 2024). However, the reward labels produced in these works are often of high variance and noisy (Ma et al., 2023a). Preference-based RL is more promising way that learns from human preferences over trajectory pairs (Christiano et al., 2017; Lee et al., 2021). On the one hand, we can learn a reward model from preferences and then optimize the policy according to the reward model (Christiano et al., 2017; Kim et al., 2023). On the other hand, the policy can be directly optimized according to the preferences (Hejna & Sadigh, 2023; Hejna et al., 2024).

However, preference-based RL requires either querying a large number of expert preference labels online (Lee et al., 2021; Park et al., 2022) or a labeled offline preference dataset (Kim et al., 2023; Hejna et al., 2024), which is quite time-consuming and expensive. Previous methods propose to use the reasoning abilities of Large Language Models (LLMs) to provide preference labels (Wang et al., 2024), but the generated labels are not guaranteed to be accurate and it is assumed to have access to the environment information.

In this paper, we propose a novel **V**ision-**L**anguage **P**reference alignment framework, named **VLP**, to provide preference feedback for video pairs given language instructions. Specifically, we collect a

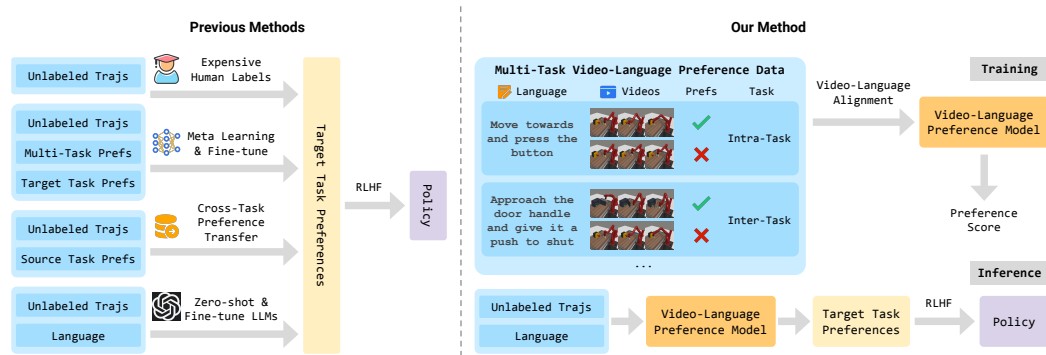

Figure 1: Comparison of VLP (right) with previous methods (left) of providing preference labels.

video dataset from various policies under augmented language instructions, which contains implicit preference relations based on the trajectory optimality and the vision-language correspondence. Then, we define language-conditioned preferences and propose a novel vision-language alignment architecture to learn a trajectory-wise preference model for preference labeling, which consists of a video encoder, a language encoder, and a cross-modal encoder to facilitate vision-language alignment. The preference model is optimized by intra-task and inter-task preferences that are implicitly contained in the dataset. In inference, VLP provides preference labels for target tasks and can even generalize to unseen tasks and unseen language instructions. We provide an analysis to show the learned preference model resembles the negative regret of the segment under mild conditions. The preference labels given by VLP are employed for various downstream preference optimization algorithms to facilitate policy learning.

In summary, our contributions are as follows: (i) We propose a novel vision-language preference alignment framework, which learns a vision-language preference model to provide preference feedback for embodied manipulation tasks. (ii) We propose language-conditioned preferences and construct a vision-language preference dataset, which contains 4800 videos with language instructions and implicit language-conditioned relations. (iii) Extensive empirical results on simulated embodied manipulation tasks demonstrate that our method provides accurate preferences and generalizes to unseen tasks and unseen language instructions, outperforming the baselines by a large margin.

## 2 BACKGROUND

**Problem Setting.** We formulate the RL problem as a Markov Decision Process (MDP) (Sutton & Barto, 2018) represented as a tuple $\mathcal{M} = (\mathcal{S}, \mathcal{A}, \mathcal{P}, \mathcal{R}, \gamma, p_0)$, where $\mathcal{S}$ is the state space, $\mathcal{A}$ is the action space, $\mathcal{P} : \mathcal{S} \times \mathcal{A} \to \mathcal{S}$ is the transition function, $\mathcal{R} : \mathcal{S} \times \mathcal{A} \to \mathbb{R}$ is the reward function, $\gamma \in [0, 1)$ is the discount factor, and $p_0 : \mathcal{S} \to [0, 1]$ is the initial state distribution. At timestep $t$, the agent observes a state $s_t$ and selects an action $a_t$ based on a policy $\pi(a_t|s_t)$. Then, the agent receives a reward $r_t$ from the environment, and the agent transits to $s_{t+1}$ according to the transition function. The agent's goal is to find a policy that maximizes the expected cumulative reward $\mathbb{E}\left[\sum_{t=0}^{\infty} \gamma^t r_t\right]$. In multi-task setting, for a task $\mathcal{T} \sim p(\mathcal{T})$, a task-specific MDP is represented as $\mathcal{M}^{\mathcal{T}} = (\mathcal{S}^{\mathcal{T}}, \mathcal{A}, \mathcal{P}^{\mathcal{T}}, \mathcal{R}^{\mathcal{T}}, \gamma, p_0^{\mathcal{T}})$.

**Preference-based RL.** Preference-based RL differs from RL in that it is assumed to have no access to the ground-truth rewards (Christiano et al., 2017; Lee et al., 2021). In preference-based RL, human teachers provide preference labels over trajectory pairs, and a reward model is learned from these preferences. Formally, a trajectory segment $\sigma$ of length $H$ is represented as $\{s_1, a_1, \dots, s_H, a_H\}$ and a segment pair is $(\sigma^1, \sigma^2)$. The preference label $y \in \{0, 1, 0.5\}$ denotes which segment is preferred, where 0 indicates $\sigma^1$ is preferred (i.e., $\sigma^1 \succ \sigma^2$), 1 indicates $\sigma^2$ is preferred (i.e., $\sigma^2 \succ \sigma^1$), and 0.5 represents two segments are equally preferred. Previous preference-based RL approaches construct a preference predictor with the reward model $\widehat{r}_\psi$ via Bradley-Terry model (Bradley & Terry, 1952):

$$P_\psi[\sigma^1 \succ \sigma^2] = \frac{\exp\left(\sum_{t=1}^{H} \widehat{r}_\psi(s_t^1, a_t^1)\right)}{\sum_{k=1}^{2} \exp\left(\sum_{t=1}^{H} \widehat{r}_\psi(s_t^k, a_t^k)\right)}, \tag{1}$$

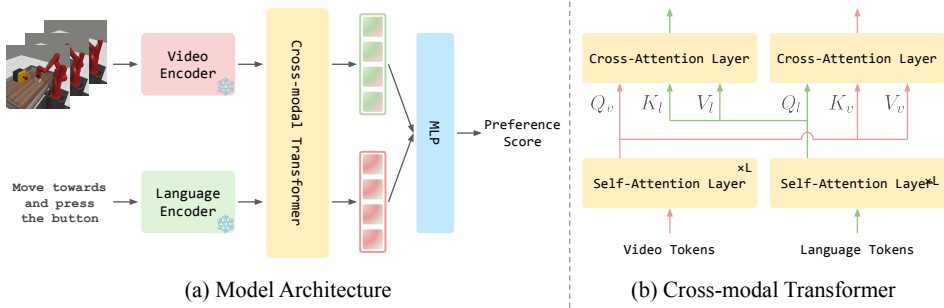

(a) Model Architecture        (b) Cross-modal Transformer

Figure 2: (a) Trajectory videos and language instruction are fed into the preference model to obtain a trajectory-wise preference score. (b) The cross-modal transformer obtains language-related video features and video-related language features by cross-attention mechanism.

where $P_\psi[\sigma^1 \succ \sigma^2]$ denotes the probability that $\sigma^1$ is preferred over $\sigma^2$ predicted by current reward model $\widehat{r}_\psi$. Assume we have a dataset with preference labels $\mathcal{D} = \{(\sigma^1, \sigma^2, y)\}$, the reward learning process can be formulated as a classification problem using cross-entropy loss (Christiano et al., 2017):

$$\mathcal{L}_{\text{ce}} = - \mathop{\mathbb{E}}_{(\sigma^1, \sigma^2, y) \sim \mathcal{D}} \Big[ (1 - y) \log P_\psi[\sigma^1 \succ \sigma^2] + y \log P_\psi[\sigma^2 \succ \sigma^1] \Big]. \tag{2}$$

By optimizing Eq. (2), the reward model is aligned with human preferences, providing reward signals for policy learning.

## 3 METHOD

In this section, we first present the overall framework of VLP, including model architecture and the vision-language preference dataset. Then, we introduce language-conditioned preferences and the detailed algorithm for vision-language preference learning, which learns a trajectory-wise preference model via vision-language preference alignment. Last, we provide a theoretical analysis of our method.

### 3.1 FRAMEWORK

The goal of VLP is to learn a generalized preference model capable of providing preferences for novel embodied tasks. To achieve this, the preference model receives videos and language as inputs, where videos serve as universal representations of agent trajectories and language act as universal and flexible instructions. To obtain high-quality representations of these two modalities, we utilize CLIP (Radford et al., 2021), which is pre-trained on extensive image-text data, as our video and language encoders. The extracted video and language features are fed into to a cross-modal transformer for cross-modal attention interaction to capture video features associated with the language and language features related to the video. These features are subsequently utilized for predicting preference scores in vision-language preference learning. The overall framework is illustrated in Figure 2.

**Model Architecture.** A video $v$ is represented as a sequence of video frames, i.e., $v = \{v_1, v_2, \ldots, v_{|v|}\}$, where $v_i \in \mathbb{R}^{H \times W \times 3}$, $H$ and $W$ are the height and width of each video frame, and $|v|$ denotes the number of video frames. The video encoder is employed to obtain the video tokens $z = \{z_1, z_2, \ldots, z_{|v|}\}$, where $z_i \in \mathbb{R}^{M \times D_v}$, $M = H/p \times W/p$ is the number of visual tokens, $p$ is the patch size of CLIP ViT, and $D_v$ is the dimension of the visual tokens. Given language input $l$, the language tokens $u \in \mathbb{R}^{N \times D_l}$ are obtained via the language encoder, where $N$ is the number of language tokens, and $D_l$ is the dimension of the language tokens.

With video tokens $z$ and language tokens $u$, a cross-modal encoder is employed to facilitate multi-modal feature learning, making tokens of different modalities fully fuse with each other. Video tokens and language tokens are separately inputted into the self-attention layers. Then, utilizing the output video tokens as queries and the output language tokens as keys and values, the cross-attention layer, as shown in Figure 2(b), generates language features that are closely related to the input video. Similarly, the cross-attention layer produces language-related video features. The multi-modal tokens are averaged along the first dimension and then concatenated as $w \in \mathbb{R}^{D_w}$, where $D_w = D_v + D_l$.

These new tokens are fed into the final Multi-layer Perceptron (MLP) for vision-language preference prediction, outputting a trajectory-level preference score.

**Vision-Language Preference Dataset.**  While there are open-sourced embodied datasets with language instructions (Mu et al., 2023), there lacks a multi-modal preference dataset for generalized preference learning. To this end, we construct MTVLP, a multi-task vision-language preference dataset built upon Meta-World (Yu et al., 2020). To that end, we consider the following aspects: (i) trajectories of various optimality levels should be collected to define clear preference relations within each task; (ii) each trajectory pair should be accompanied with a corresponding language instruction for learning language-conditioned preferences.

It is easy to describe the optimality of expert trajectories and random trajectories because it is easy to understand the agent's behavior in these trajectories. However, it is challenging to define a medium-level policy without explicit rewards. Fortunately, we find most robot tasks can be divided into multiple stages, where each stage completes a part of the overall task. Thus, we define a medium-level policy as successfully completing half of the stages of the task. For example, we divided the task of *opening the drawer* into two subtasks: (i) moving and grasping the drawer handle and (ii) pulling the drawer handle. A medium-level policy only completes the first subtask.

We leverage a scripted policy for each task to roll out trajectories of three optimality levels: expert, medium, and random. For expert-level trajectories, we employ the scripted policy with Gaussian noise to interact. The medium-level trajectories are also collected with the scripted policy but are terminated when the half of subtasks are completed. As for random-level trajectories, actions are randomly sampled from a uniform distribution during rollout. For the corresponding language, we obtain diverse language instructions to improve the generalization abilities of our model by aligning one video with multiple similar language instructions. Following (Adeniji et al., 2023), we query GPT-4V (OpenAI, 2023) to generate language instructions with various verb structure examples and synonym nouns of each task. Details of collecting trajectories and language instructions for each task are shown in Appendix B.

## 3.2 VISION-LANGUAGE PREFERENCE ALIGNMENT

**Language-conditioned Preferences.**  Previous RLHF methods define trajectory preferences according to a single task goal. However, this uni-modal approach struggles to generalize to new tasks due to its rigid preference definition. In contrast, by integrating language as a condition, we can establish more flexible preference definitions. Consider two videos, $v_1^1$ and $v_2^1$, along with a language instruction $l^1$ from task $\mathcal{T}^1$, and another video $v^2$ paired with a language instruction $l^2$ from task $\mathcal{T}^2$. We categorize three forms of language-conditioned preferences: Intra-Task Preference (ITP), Inter-Language Preference (ILP), and Inter-Video Preference (IVP), as shown in Table 1.

Table 1: Three types of language-conditioned preferences.

| Preference Type | Videos | Language | Criterion |
|---|---|---|---|
| Intra-Task Preference | $v_1^1, v_2^1 \sim \mathcal{T}^1$ | $l^1 \sim \mathcal{T}^1$ | optimality |
| Inter-Language Preference | $v_1^1, v_2^1 \sim \mathcal{T}^1$ | $l^2 \sim \mathcal{T}^2$ | equally preferred |
| Inter-Video Preference | $v_1^1 \sim \mathcal{T}^1, v_1^2 \sim \mathcal{T}^2$ | $l^1 \sim \mathcal{T}^1$ | $v_1^1 \succ v_2^1 \| l^1$ |

ITP corresponds to the conventional case of preference relation within the same task (Christiano et al., 2017), where the videos and language instructions are from the same task, and the preference relies on the optimality of videos w.r.t. the task objective. ILP considers a scenario where the language instruction differs from the task of the videos. Thus, both videos are equally preferred under this language condition. IVP deals with preferences of two videos from different tasks, with the language instruction from either task. It is straightforward to define the preference that the vision-language come from the same task is preferred to the other pair.

This framework allows for the establishment of universal and adaptable preference relations, wherein videos from the same task can yield varying preference labels depending on the language condition. Notably, even random trajectories paired with language instructions from a specific task is preferred to expert trajectories from other tasks.

**Vision-Language Preference Learning.** With language-conditioned preferences defined above, we further introduce our vision-language preference learning algorithm. We aim to develop a vision-language preference model that predicts the preferred video under specific language conditions. However, directly inputting two videos and a language instruction into the model would affect computational efficiency. So, we consider the conventional way to learn from preference labels (Christiano et al., 2017), i.e., first constructing preference predictors via Bradley-Terry model (Bradley & Terry, 1952). Previous work has revealed the advantages of learning a preference model over a reward model (Zhang et al., 2024). Based on these insights, our proposed preference model $f_\psi(v|l)$ takes a video and a language instruction as inputs and outputs a scalar preference score. Then the preference label can be obtained by comparing preference scores of two videos with a given language instruction, i.e., $v_1 \succ v_2|l$ if $f_\psi(v_1|l) > f_\psi(v_2|l)$.

Given videos $v_1$ representing $\sigma_1$ and $v_2$ representing $\sigma_2$, the language-conditioned preference distribution $P_\psi[v_1 \succ v_2|l]$ is the probability that $\sigma_1$ is preferred over $\sigma_2$ under the condition $l$:

$$P_\psi[v_1 \succ v_2|l] = \frac{\exp\big(f_\psi(v_1|l)\big)}{\exp\big(f_\psi(v_1|l)\big) + \exp\big(f_\psi(v_2|l)\big)}. \quad (3)$$

Given tasks $\mathcal{T}^1$ and $\mathcal{T}^2$, we consider the following objectives aligned with language-conditioned preference relations: **(a)** Learning Intra-Task Preference: Within the same task, the video that better follows $l$ should be preferred, analogous to previous RLHF objective (Christiano et al., 2017); **(b)** Learning Inter-Language Preference: Under the language condition of task $\mathcal{T}^2$, videos from task $\mathcal{T}^1$ are equally preferred; **(c)** Learning Inter-Video Preference: Under the language condition of task $\mathcal{T}^1$, the video from $\mathcal{T}^1$ is preferred over the video from $\mathcal{T}^2$.

During vision-language preference learning, a task $\mathcal{T}$ is sampled from all training tasks, followed by sampling a minibatch $\{v_1^b, v_2^b, v^{\neq b}, l^b, l^{\neq b}, y^{\text{ITP}}, y^{\text{ILP}}, y^{\text{IVP}}\}^{1:B}$. Here, the superscript $^b$ indicates data sampled from task $\mathcal{T}$ in the minibatch, while $^{\neq b}$ denotes data from other tasks. $y^{\text{ITP}}, y^{\text{ILP}}, y^{\text{IVP}}$ are the ground-truth labels of ITP, ILP, and IVP, respectively. The total loss of vision-language preference learning is as follows:

$$\mathcal{L}_{\text{ce}} = -\sum_{b \in B} \Big[ \underbrace{\text{CE}\big(P_\psi[v_1^b \succ v_2^b|l^b], y^{\text{ITP}}\big)}_{(a)} + \lambda_1 \underbrace{\text{CE}\big(P_\psi[v_1^b \succ v_2^b|l^{\neq b}], y^{\text{ILP}}\big)}_{(b)}$$
$$+ \lambda_2 \underbrace{\text{CE}\big(P_\psi[v_1^b \succ v^{\neq b}|l^b], y^{\text{IVP}}\big)}_{(c)} + \lambda_2 \underbrace{\text{CE}\big(P_\psi[v_2^b \succ v^{\neq b}|l^b], y^{\text{IVP}}\big)}_{(c)} \Big], \quad (4)$$

where $\text{CE}(\cdot, \cdot)$ is the cross-entropy loss, and $\lambda_1$ and $\lambda_2$ are balance weights of learning ILP and IVP. By optimizing Eq. (4), the vision-language preference model outputs trajectory-level preference scores aligned with the language-conditioned preference relations.

### 3.3 THEORETICAL ANALYSIS

Considering an MDP with a reward function $r$, we define $Q_r^*$ and $V_r^*$ as the state-action value function and state value function for an optimal policy $\pi^*$ under $r$. Then if a segment $\sigma$ is optimal with respect to the reward function $r$, we have $Q_r^*(s_t^\sigma, a_t^\sigma) = V_r^*(s_t^\sigma)$. In this case, we have $A_r^*(s_t^\sigma, a_t^\sigma) = 0$, and the *regret* is 0. Formally, we define the regret for a full segment as

$$\text{regret}(\sigma|r) = \sum_{t=1}^H \text{regret}(\sigma_t|r) = \sum_{t=1}^H \Big[V_r^*(s_t^\sigma) - Q_r^*(s_t^\sigma, a_t^\sigma)\Big] = \sum_{t=1}^H -A_r^*(s_t^\sigma, a_t^\sigma). \quad (5)$$

The regret $\text{regret}(\sigma|r) > 0$ holds for all sub-optimal segments. The following proposition shows that the proposed video-based preference model in Eq. (3) recovers the regret preference model.

**Proposition 3.1.** *Under mild conditions, the learned preference distribution $P_\psi$ approximates the regret preference distribution as:*

$$P_{\text{regret}}^*[\sigma^1 \succ \sigma^2|r] = \frac{\exp\big(-\text{regret}(\sigma^1|r)\big)}{\exp\big(-\text{regret}(\sigma^1|r)\big) + \exp\big(-\text{regret}(\sigma^2|r)\big)}, \quad (6)$$

*and the preference model $f_\psi(\sigma|l)$ is learned to approximate $-\text{regret}(\sigma|r) + c$, where $c$ is a constant.*

We give a detailed analysis in Appendix A. In practice, we cannot obtain the true reward function but acquire the given task's language instruction $l$. We hypothesize the reward-related information is implicitly included in the language instruction $l$ and propose several pseudo-labels $\{y^{\text{ITP}}, y^{\text{ILP}}, y^{\text{IVP}}\}$ according to the optimality of segment and segment-language correspondence. The use of language in preference model provides generalization ability of preferences among language instructions. Meanwhile, we adopt video $v$ to represent segment $\sigma$ and propose a novel architecture for vision-language alignment.

# 4 RELATED WORK

**Vision-Language Models for Reinforcement Learning.**    Our work is related to the literature on VLM rewards and preferences for embodied manipulation tasks (Radford et al., 2021; Nair et al., 2023; Ma et al., 2023a; Rocamonde et al., 2024; Wang et al., 2024; Liu et al., 2024a). These methods can be divided into three categories: (i) representation-based pre-training, (ii) zero-shot inference, and (iii) downstream fine-tuning. For representation-based approaches, R3M (Nair et al., 2023) is pre-trained on the Ego4D dataset (Grauman et al., 2022) to learn useful representations for downstream tasks. LIV (Ma et al., 2023b), which extends VIP (Ma et al., 2023b) to multi-modal representations, is pre-trained on EpicKitchen dataset (Damen et al., 2018), and can also be fine-tuned on target domain. For zero-shot inference methods, VLM-RM (Rocamonde et al., 2024) utilizes CLIP (Radford et al., 2021) as zero-shot vision-language rewards. RoboCLIP (Sontakke et al., 2023) uses S3D (Xie et al., 2018), which is pre-trained on HowTo100M dataset (Miech et al., 2019), as video-language model to compute vision-language reward with a single demonstration (a video or a text). RL-VLM-F (Wang et al., 2024) leverages Gemini-Pro (Team et al., 2023) and GPT-4V (OpenAI, 2023) for zero-shot preference feedback. CriticGPT (Liu et al., 2024a) is the representative method of (iii), which fine-tunes multimodal LLMs on a instruction-following dataset, and utilizes the tuned model to provide preference feedback for downstream policy learning. VLP differs from these approaches that we do not suffer from burdensome training of (i) and (iii), showing great computing efficiency. And VLP learns more embodied manipulation knowledge compared with VLMs pre-trained on natural image-text data.

**Preference-based Reinforcement Learning.**    Preference-based RL is a promising framework for aligning the agent with human values. However, feedback efficiency is a crucial challenge in preference-based RL, with multiple recent studies striving to tackle. PEBBLE (Lee et al., 2021) improves the efficiency by unsupervised pre-training. SURF (Park et al., 2022) proposes to obtain pseudo labels using reward confidence. RUNE (Liang et al., 2022) employs reward uncertainty to guide exploration. Meta-Reward-Net (Liu et al., 2022) takes advantage of the performance of the Q-function as an additional signal to refine the accuracy of the reward model. Hejna III & Sadigh (2023) leverages meta-learning to pre-train the reward model, enabling fast adaptation to new tasks with few preference labels. Recently, a growing number of studies focus on offline preference-based RL with the population of offline RL (Levine et al., 2020; Kostrikov et al., 2022; Lyu et al., 2024). PT (Kim et al., 2023) introduces a Transformer-based architecture for reward modeling. OPPO (Kang et al., 2023) proposes to learn policies without a reward function. IPL (Hejna & Sadigh, 2023) learns the Q-function from preferences, also eliminating the need of reward learning. CPL (Hejna et al., 2024) further views preference-based RL as a supervised learning problem, directly learning policies from preferences. FTB (Zhang et al., 2024) introduces a diffusion model for better trajectory generation. PEARL (Liu et al., 2024b) proposes cross-task preference alignment to transfer preference labels between tasks and learn reward models robustly via reward distributional modeling. VLP addresses the labeling cost by learning a vision-language preference model via vision-language alignment, thereby providing generalized preferences to novel tasks.

# 5 EXPERIMENTS

In this section, we evaluate VLP on Meta-World (Yu et al., 2020) benchmark and aim to answer the following questions:

- **Q1:** How do VLP labels compare with scripted labels in offline RLHF? (Section 5.2)
- **Q2:** How does VLP compare with other vision-language rewards approaches? (Section 5.3)

Table 2: Success rate of RLHF methods with scripted labels and VLP labels. The results are reported with mean and standard deviation across five random seeds. The result of VLP is shaded and is **bolded** if it exceeds or is comparable with that of RLHF approaches with scripted labels. VLP Acc. denotes the accuracy of preference labels inferred by VLP compared with scripted labels.

| Task | P-IQL | | | IPL | | | CPL | | | VLP Acc. | |
|---|---|---|---|---|---|---|---|---|---|---|---|
| | **Human** | **Scripted** | **VLP** | **Human** | **Scripted** | **VLP** | **Human** | **Scripted** | **VLP** | **Human** | **Scripted** |
| Button Press | $93.1_{\pm 5.2}$ | $72.6_{\pm 7.1}$ | $\mathbf{90.1}_{\pm 3.9}$ | $65.2_{\pm 7.2}$ | $50.6_{\pm 7.9}$ | $\mathbf{56.0}_{\pm 1.4}$ | $85.0_{\pm 7.2}$ | $74.5_{\pm 8.2}$ | $\mathbf{83.9}_{\pm 11.8}$ | 99.0 | 93.0 |
| Door Close | $79.2_{\pm 6.3}$ | $79.2_{\pm 6.3}$ | $\mathbf{79.2}_{\pm 6.3}$ | $61.5_{\pm 9.4}$ | $61.5_{\pm 9.4}$ | $\mathbf{61.5}_{\pm 9.4}$ | $98.5_{\pm 1.0}$ | $98.5_{\pm 1.0}$ | $\mathbf{98.5}_{\pm 1.0}$ | 100.0 | 100.0 |
| Drawer Close | $63.7_{\pm 6.4}$ | $49.3_{\pm 4.2}$ | $\mathbf{64.9}_{\pm 2.9}$ | $63.2_{\pm 4.6}$ | $64.3_{\pm 9.6}$ | $63.2_{\pm 4.7}$ | $54.1_{\pm 8.7}$ | $45.6_{\pm 3.5}$ | $\mathbf{57.5}_{\pm 14.3}$ | 96.0 | 96.0 |
| Faucet Close | $51.1_{\pm 7.5}$ | $51.1_{\pm 7.5}$ | $\mathbf{51.1}_{\pm 7.5}$ | $45.4_{\pm 8.6}$ | $45.4_{\pm 8.6}$ | $\mathbf{45.4}_{\pm 8.6}$ | $80.0_{\pm 2.9}$ | $80.0_{\pm 2.9}$ | $\mathbf{80.0}_{\pm 2.9}$ | 100.0 | 100.0 |
| Window Open | $69.7_{\pm 6.8}$ | $62.4_{\pm 6.4}$ | $\mathbf{69.7}_{\pm 6.8}$ | $61.4_{\pm 8.6}$ | $54.1_{\pm 6.7}$ | $\mathbf{61.4}_{\pm 8.6}$ | $99.1_{\pm 1.1}$ | $91.6_{\pm 1.7}$ | $\mathbf{99.1}_{\pm 1.1}$ | 100.0 | 98.0 |
| **Average** | 71.4 | 62.9 | **71.0** | 59.3 | 55.2 | **57.5** | 83.3 | 78.0 | **83.8** | 99.0 | 97.4 |

- **Q3:** How does VLP generalize to unseen tasks and language instructions? (Section 5.4)
- **Q4:** How do hyperparameters influence VLP? (Section 5.5)

## 5.1 SETUP

**Implementation Details.** We evaluate VLP on the 5 test tasks of MTVLP, including *Button Press*, *Door Close*, *Drawer Close*, *Faucet Close*, and *Window Open*, while the other 45 tasks of Meta-World (Yu et al., 2020) are used as training tasks. For implementing VLP, we use the pre-trained ViT-B/16 CLIP model (Radford et al., 2021) as our video encoder and language encoder. The weights of learning ILP and IVP in Eq. (4) are $\lambda_1 = 0.1$, $\lambda_2 = 0.5$, respectively. Additional hyperparameters of VLP are detailed in Table 10 in Appendix C. All experiments are conducted on a single NVIDIA RTX 4090 GPU.

## 5.2 HOW DO VLP LABELS COMPARE WITH SCRIPTED LABELS IN OFFLINE RLHF?

**Baselines.** We evaluate VLP by combining it with recent offline RLHF algorithms: (i) **P-IQL** (Preference IQL), which first learns a reward model from preferences and then learns a policy via IQL (Kostrikov et al., 2022); (ii) **IPL** (Hejna & Sadigh, 2023), which learns a policy without reward learning by aligning the Q-function with preferences; (iii) **CPL** (Hejna et al., 2024), which directly learns a policy using a contrastive objective with maximum entropy principle, eliminating the need for reward learning and RL.

**Evaluation.** For each evaluation task, we train each RLHF method with scripted labels (Christiano et al., 2017; Lee et al., 2021) and VLP labels (denoted as **+VLP**), respectively. Scripted preference labels mean the preference labels computed based on the ground-truth rewards (Christiano et al., 2017; Lee et al., 2021). The number of preference labels is set to 100 for all tasks. The evaluation is conducted over 25 episodes every 5000 steps. Following (Hejna et al., 2024), we average the results of 8 neighboring evaluations and take the maximum value among all averaged values as the result. Detailed hyperparameters of RLHF algorithms can be found in Appendix C.

**Results.** Experimental results in Table 2 demonstrate that the performance of P-IQL+VLP and CPL+VLP is comparable with, and in some cases, outperforms that with scripted labels on all evaluation tasks. We hypothesize that the ground-truth reward of *Button Press, Drawer Close* and *Window Open* may not accurately represent the task goal (Ma et al., 2024; Xie et al., 2024). However, by aligning video and language modalities through preference relations with language as conditions, the predicted VLP labels directly represent how the video reflects the language instruction. Therefore, our method provides more accurate and preference labels and can generalize to unseen tasks.

## 5.3 HOW DOES VLP COMPARE WITH OTHER VISION-LANGUAGE REWARDS APPROACHES?

**Baselines.** We compare VLP with the following VLM rewards baselines: (i) **R3M** (Nair et al., 2023), which pre-trains visual representation by time-contrastive learning and vision-language alignment; (ii) **VIP** (Ma et al., 2023b), which provides generalized visual reward and representation for downstream tasks via value-implicit pre-training; (iii) **LIV** (Ma et al., 2023a), which learns

Table 3: Success rate of VLP (i.e., P-IQL trained with VLP labels) against IQL with VLM **rewards**. The results are reported with mean and standard deviation across five random seeds. The result of VLP is  shaded  and the best score of all methods is **bolded**.

| Task | R3M | VIP | LIV | CLIP | VLM-RM (0.0) | VLM-RM (1.0) | VLP |
|------|-----|-----|-----|------|--------------|--------------|-----|
| Button Press | $10.1 \pm 2.3$ | $68.4 \pm 6.4$ | $56.3 \pm 1.9$ | $59.5 \pm 6.1$ | $60.3 \pm 6.1$ | $64.3 \pm 8.4$ | $\mathbf{90.1} \pm 3.9$ |
| Door Close | $70.9 \pm 5.3$ | $74.8 \pm 9.5$ | $43.3 \pm 3.2$ | $43.6 \pm 3.9$ | $45.8 \pm 8.5$ | $41.1 \pm 3.4$ | $\mathbf{79.2} \pm 6.3$ |
| Drawer Close | $46.6 \pm 2.6$ | $70.4 \pm 4.5$ | $61.8 \pm 5.7$ | $69.4 \pm 4.1$ | $69.4 \pm 4.5$ | $\mathbf{73.5} \pm 5.4$ | $64.9 \pm 2.9$ |
| Faucet Close | $25.7 \pm 23.6$ | $40.9 \pm 8.0$ | $42.2 \pm 6.3$ | $59.6 \pm 7.5$ | $\mathbf{60.1} \pm 5.1$ | $33.7 \pm 15.3$ | $51.1 \pm 7.5$ |
| Window Open | $39.0 \pm 6.6$ | $42.7 \pm 11.3$ | $33.8 \pm 6.4$ | $26.4 \pm 2.0$ | $23.9 \pm 1.9$ | $23.7 \pm 4.9$ | $\mathbf{69.7} \pm 6.8$ |
| **Average** | 38.5 | 59.4 | 47.5 | 51.7 | 51.9 | 47.3 | **71.0** |

Table 4: Success rate of VLP (i.e., P-IQL trained with VLP labels) against P-IQL with VLM **preferences** (denoted with prefix **P-**). The results are reported with mean and standard deviation across five random seeds. The result of VLP is  shaded  and the best score of all methods is **bolded**.

| Task | P-R3M | P-VIP | P-LIV | P-CLIP | P-VLM-RM (0.0) | P-VLM-RM (1.0) | RoboCLIP | VLP |
|------|-------|-------|-------|--------|----------------|----------------|----------|-----|
| Button Press | $84.7 \pm 5.8$ | $41.2 \pm 3.9$ | $61.7 \pm 5.1$ | $62.9 \pm 6.2$ | $72.8 \pm 5.0$ | $44.2 \pm 4.2$ | $56.4 \pm 7.3$ | $\mathbf{90.1} \pm 3.9$ |
| Door Close | $72.4 \pm 11.5$ | $54.2 \pm 13.8$ | $67.9 \pm 6.3$ | $53.3 \pm 10.3$ | $57.6 \pm 2.9$ | $45.7 \pm 7.6$ | $47.6 \pm 6.7$ | $\mathbf{79.2} \pm 6.3$ |
| Drawer Close | $59.6 \pm 6.5$ | $63.0 \pm 3.7$ | $45.5 \pm 10.4$ | $63.4 \pm 3.2$ | $62.7 \pm 3.0$ | $49.2 \pm 6.9$ | $\mathbf{73.0} \pm 6.2$ | $64.9 \pm 2.9$ |
| Faucet Close | $58.0 \pm 4.5$ | $51.1 \pm 7.5$ | $\mathbf{62.3} \pm 7.2$ | $60.2 \pm 10.4$ | $57.3 \pm 7.0$ | $51.3 \pm 9.5$ | $62.1 \pm 6.3$ | $51.1 \pm 7.5$ |
| Window Open | $27.3 \pm 5.0$ | $50.2 \pm 1.8$ | $22.2 \pm 18.1$ | $28.4 \pm 3.2$ | $33.2 \pm 5.4$ | $20.7 \pm 2.3$ | $28.1 \pm 4.6$ | $\mathbf{69.7} \pm 6.8$ |
| **Average** | 60.4 | 51.9 | 51.9 | 53.6 | 56.7 | 42.2 | 53.4 | **71.0** |

vision-language rewards and representation via multi-modal value pre-training; (iv) **CLIP** (Radford et al., 2021), which pre-trains by aligning vision-language representation on a large-scale image-text pairs dataset; (v) **VLM-RM** (Rocamonde et al., 2024), which provides zero-shot VLM rewards based on CLIP (Radford et al., 2021). VLM-RM includes a hyperparameter $\alpha$, which controls the goal-baseline regularization strength. In the evaluation, we denote the variant of $\alpha = 0.0$ as **VLM-RM (0.0)** and the variant of $\alpha = 1.0$ as **VLM-RM (1.0)**. (vi) **RoboCLIP** (Sontakke et al., 2023), which provides zero-shot VLM rewards using pre-trained video-language models and a single demonstration (a video demonstration or a language description) of the task.

**Evaluation.** We first evaluate our method with the VLM baselines by directly training IQL with VLM **rewards**. VLP is tested by training P-IQL with VLP labels, and the experimental setting of our method is the same as that of Section 5.2. We further compare VLP with VLM **preferences**, i.e., using predicted VLM rewards to compute preference labels for a fair comparison with our method. However, RoboCLIP obtains scalar trajectory-level rewards and we utilize them as trajectory return for preference labels calculation. Implementation details of IQL and VLM baselines can be found in Appendix C.

**Results.** Results in Table 3 show that our method exceeds the VLM baselines that train IQL from VLM rewards by a large margin with an average success rate of **71.0**. As shown in Table 4, when the VLM baselines are trained with preferences computed by VLM rewards, our method still surpasses the baselines. We further compute the preference label accuracy of each method, detailed in Table 15. The results show that VLP exceeds VLM baselines, which do not learn relative relations of reward values.

**Reward / Preference Correlation.** To further investigate the advantages of VLP model compared with VLM reward models, we compare the correlation between VLM rewards with ground-truth rewards and VLP labels with scripted preference labels. Results in Table 5 indicate that VLP labels exhibit a stronger correlation with scripted labels compared with VLM rewards.

## 5.4 HOW DOES VLP GENERALIZE TO UNSEEN TASKS AND LANGUAGE INSTRUCTIONS?

**Evaluation.** We first evaluate how accurate 3 kinds of VLP labels are on the test tasks. We test the preference model with phrases, descriptions, and correct and incorrect object colors. Since the label of ILP is $0.5$ (i.e., two segments are equally preferred), we compute ILP loss with the (b) term in

Table 5: The correlation coefficient of VLM rewards with ground-truth rewards and VLP labels with scripted preference labels. Larger correlation means the predicted values are more correlated with the ground-truth.

| Task | R3M | VIP | LIV | CLIP | VLM-RM (0.0) | VLM-RM (1.0) | VLP |
|------|-----|-----|-----|------|-------------|-------------|-----|
| Button Press | 0.313 | 0.204 | -0.281 | 0.127 | 0.153 | -0.082 | **0.581** |
| Door Close | 0.735 | 0.125 | 0.600 | -0.309 | -0.152 | -0.492 | **1.000** |
| Drawer Close | -0.106 | 0.043 | 0.052 | -0.151 | -0.137 | -0.031 | **0.438** |
| Faucet Close | 0.676 | 0.851 | 0.563 | -0.301 | -0.291 | 0.084 | **1.000** |
| Window Open | 0.411 | **0.725** | -0.568 | 0.336 | 0.405 | -0.333 | 0.571 |
| **Average** | 0.406 | 0.390 | 0.073 | -0.060 | -0.005 | -0.171 | **0.718** |

Eq. (4), i.e., $-\sum_{b\in B} \mathrm{CE}\left(P_\psi[v_1^b \succ v_2^b | l^{\neq b}], y^{\mathrm{ILP}}\right)$. Performance of ITP and IVP are measured with accuracy. Experimental details can be found in Appendix C.

**Results.** Table 6 shows that VLP generalizes to unseen language instructions on unseen tasks with high ITP and IVP accuracy and low ILP loss. However, using unseen phrases as language conditions leads to a performance drop, while unseen descriptions have a slight negative impact on ITP but a positive impact on IVP and ILP. We think the reason is that phrases contain insufficient information about completing the task, while descriptions contain enough task information. VLP generalizes well with suitable language information of tasks. Also, VLP exhibits strong generalization abilities on color.

Table 6: The generalization abilities of our method on 5 unseen tasks with different types of language instructions. Acc. denotes the accuracy of preference labels inferred by VLP compared with ground-truth labels.

| Metric | Seen | Phrase | Description | Correct Color | Incorrect Color |
|--------|------|--------|-------------|---------------|-----------------|
| **ITP Acc. (↑)** | 97.4 | 95.8 | 97.0 | 97.0 | 97.0 |
| **IVP Acc. (↑)** | 91.7 | 90.5 | 91.9 | 91.9 | 91.8 |
| **ILP Loss (↓)** | 0.705 | 0.704 | 0.704 | 0.705 | 0.705 |
| **Average Loss (↓)** | 0.555 | 0.554 | 0.558 | 0.556 | 0.557 |

## 5.5 ABLATION STUDIES

**Attention Map Visualization.** We further analyze VLP by visualizing the attention maps of the cross-attention. Results in Figure 3 show that regions of the objects related to language instructions exhibit high attention weights. For example, in the Drawer Close task, our vision-language preference model specifically focuses on whether the drawer is closed, with the attention map highlighting the edges of the drawer to monitor its position and similarly for Door Close task. These observations demonstrate that our vision-language preference model effectively learns to guide language tokens to attend to relevant regions in the videos and illustrate the effectiveness of our cross-attention mechanism in bridging vision and language modalities for precise task understanding.

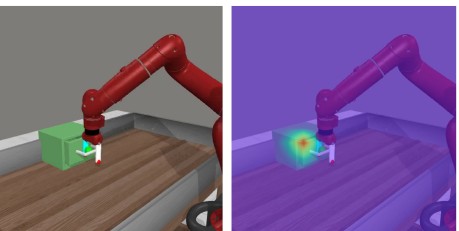 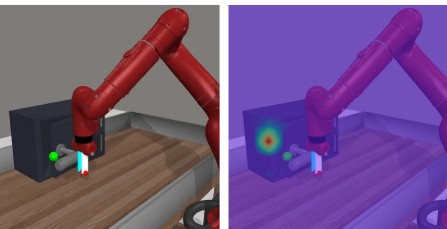

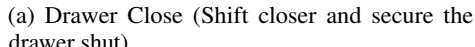

(a) Drawer Close (Shift closer and secure the drawer shut)

(b) Door Close (Direct the gripper to the door handle and press to seal it)

Figure 3: Attention maps visualization of *Drawer Close* and *Door Close*. The language instruction is shown at the bottom of each subfigure.

**Effects of $\lambda_1$ and $\lambda_2$.** $\lambda_1$ and $\lambda_2$ in Eq. (4) control the strength of ILP and IVP learning, respectively. To investigate how $\lambda_1$ and $\lambda_2$ influence VLP, we conduct experiments by vary $\lambda_1$ across $\{0.0, 0.1, 0.5\}$ and $\lambda_2$ across $\{0.0, 0.5, 1.0\}$. Results in Table 7 show that the performance of VLP drops with too small or too large $\lambda_1$. Meanwhile, without IVP learning (i.e., $\lambda_2 = 0$), the performance of IVP and ILP significantly decreases. We speculate that IVP is crucial for language-conditioned preference learning. Without IVP learning, the learned VLP model degenerates into a vanilla preference model without language as conditions.

Table 7: Accuracy of VLP labels with different loss. Acc. denotes the accuracy of preference labels inferred by VLP compared with ground-truth labels.

| $\lambda_1$ | $\lambda_2$ | ITP Acc. ($\uparrow$) | IVP Acc. ($\uparrow$) | ILP Loss ($\downarrow$) | Average Loss ($\downarrow$) |
|---|---|---|---|---|---|
| 0.0 | 0.5 | 95.4 | 74.1 | 0.728 | 0.618 |
| 0.5 | 0.5 | 85.8 | 74.7 | 0.702 | 0.578 |
| 0.1 | 0.0 | 96.2 | 63.0 | 0.775 | 0.646 |
| 0.1 | 1.0 | 95.8 | 96.5 | 0.699 | 0.554 |
| 0.1 | 0.5 | 97.4 | 91.7 | 0.705 | 0.555 |

**Effects of Preference Dataset Size.** We investigate how the preference dataset size influences our method. We conduct additional experiments by varying the dataset size across $\{50\%, 75\%, 100\%\}$. Results in Table 8 indicate that the performance of VLP downgrades as the dataset size decreases.

Table 8: Accuracy of VLP labels with different data size. Acc. denotes the accuracy of preference labels inferred by VLP compared with ground-truth labels.

| Data Size | ITP Acc. ($\uparrow$) | IVP Acc. ($\uparrow$) | ILP Loss ($\downarrow$) | Average Loss ($\downarrow$) |
|---|---|---|---|---|
| 50% | 94.2 | 89.6 | 0.699 | 0.557 |
| 75% | 95.2 | 89.7 | 0.707 | 0.555 |
| 100% | 97.4 | 91.7 | 0.705 | 0.555 |

## 6 CONCLUSION

In this paper, we propose VLP, a novel vision-language preference learning framework providing generalized preference feedback for embodied manipulation tasks. In our framework, we learn a vision-language preference model via proposed language-conditioned preference relations from the collected vision-language preference dataset. Experimental results on multiple simulated robotic manipulation tasks demonstrate that our method exceeds previous VLM rewards approaches and predicts accurate preferences compared with scripted labels. The results also show our method generalizes well to unseen tasks and unseen language instructions.

**Limitations** In this paper, we focus on providing preferences for robotic manipulation tasks. First, VLP is limited to the tasks that can be specified via videos and language instructions. While this covers a wide range of robotic tasks, certain tasks cannot be fully expressed via videos and language, such as complex assembly tasks requiring intricate spatial reasoning. Consequently, the risk of predicting incorrect preferences grows for complex tasks that are difficult to express. Second, if the language instruction lacks sufficient information of the task goal, the risk of giving incorrect labels still grows, as shown in Table 6. Last, the theoretical analysis assumes access to all possible segments, which introduces a gap between the theoretical guarantees and empirical applications.

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

# A  Theoretical Analysis

In this section, we give a theoretical analysis of the preference model $f_\psi(\sigma|l)$. The segment $\sigma$ contains $H$ states and actions as $(s_1^\sigma, a_1^\sigma, \ldots, s_H^\sigma, a_H^\sigma)$ without reward information. Meanwhile, we denote step-wise transition as $\sigma_t = (s_t^\sigma, a_t^\sigma, s_{t+1}^\sigma)$. For any reward function $r$, we define $Q_r^*$ and $V_r^*$ as the state-action value function and state value function for an optimal policy $\pi^*$ under $r$. Then, the optimal advantage function is defined as $A_r^*(s, a) \triangleq Q_r^*(s, a) - V_r^*(s)$.

Given above definitions, if a segment $\sigma$ is optimal with respect to the reward function $r$, we have $Q_r^*(s_t^\sigma, a_t^\sigma) = V_r^*(s_t^\sigma)$. In this case, we have $A_r^*(s_t^\sigma, a_t^\sigma) = 0$, and the *regret* is 0. In other cases, the regret will be greater than 0. For a segment $\sigma$, we define the single-step regret as

$$\mathrm{regret}(\sigma_t|r) = V_r^*(s_t^\sigma) - Q_r^*(s_t^\sigma, a_t^\sigma) = -A_r^*(s_t^\sigma, a_t^\sigma), \tag{7}$$

Then, the *regret* for a full segment is

$$\mathrm{regret}(\sigma|r) = \sum_{t=1}^H \mathrm{regret}(\sigma_t|r) = \sum_{t=1}^H \left[ V_r^*(s_t^\sigma) - Q_r^*(s_t^\sigma, a_t^\sigma) \right] = \sum_{t=1}^H -A_r^*(s_t^\sigma, a_t^\sigma). \tag{8}$$

According to Eq. (8), the regret value $\mathrm{regret}(\sigma|r) > 0$ holds for all sub-optimal segments, and $\mathrm{regret}(\sigma|r) = 0$ for the optimal segments with respect to the reward function $r$. Then, the regret preference distribution is defined as:

$$P_{\mathrm{regret}}^*[\sigma^1 \succ \sigma^2|r] = \frac{\exp\left(\sum_{\sigma^1} A_r^*(s_t^{\sigma^1}, a_t^{\sigma^1})\right)}{\exp\left(\sum_{\sigma^1} A_r^*(s_t^{\sigma^1}, a_t^{\sigma^1})\right) + \exp\left(\sum_{\sigma^2} A_r^*(s_t^{\sigma^2}, a_t^{\sigma^2})\right)} \tag{9}$$

$$= \frac{\exp\left(-\mathrm{regret}(\sigma^1|r)\right)}{\exp\left(-\mathrm{regret}(\sigma^1|r)\right) + \exp\left(-\mathrm{regret}(\sigma^2|r)\right)}. \tag{10}$$

We remark that such a regret preference distribution has been adopted in CPL (Hejna et al., 2024). However, CPL considers single-step regret (i.e., $-A_r^*(s_t^\sigma, a_t^\sigma)$) and adopts a maximum entropy RL framework to map the advantage function to the policy via $A_r^*(s, a) = \alpha \log \pi^*(a|s)$. In contrast, our method considers the regret of a segment as a whole and tries to model the regret of a segment directly. Recall the preference distribution in our method is defined in Eq. (3), as

$$P_\psi[v_1 \succ v_2|l] = \frac{\exp\left(f_\psi(v_1|l)\right)}{\exp\left(f_\psi(v_1|l)\right) + \exp\left(f_\psi(v_2|l)\right)}. \tag{11}$$

According to Eq. (10) and Eq. (11), the proposed preference model $f_\psi$ can be considered as *parameterized negative regret* that approximates the *true negative regret* of the whole segment. In the following, we analyze the two main difficulties in approximating $P_{\mathrm{regret}}^*(\sigma^1 \succ \sigma^2|r)$ through $P_\psi[v_1 \succ v_2|l]$.

(i) $P_{\mathrm{regret}}^*$ is calculated based on the preference data $\{(\sigma^1, \sigma^2, y)\}$ based on the regret value in Eq. (8), which is calculated based on the ground truth reward function $r(s, a)$. However, we cannot empirically obtain the true reward function but acquire the given task's language instruction $l$. We hypothesize the reward-related information is implicitly included in the language $l$ and propose several pseudo-labels $\tilde{y} \in \{y^{\mathrm{ITP}}, y^{\mathrm{ILP}}, y^{\mathrm{IVP}}\}$ according to the optimality of segment and segment-language correspondence. The use of pseudo-preference labels reduces the reliance on the ground-truth reward function and the preference distribution $P_\psi$ can be learned from implicit preference data $\{(\sigma^1, \sigma^2, \tilde{y})\}$.

(ii) Modeling the regrets of segments rather than step-wise rewards as in previous RLHF methods (Lambert et al., 2024) is more difficult. To this end, we propose a novel architecture that adopts a video encoder to obtain the video tokens and mean pooling to obtain a video representation. Then, a cross-modal encoder is employed to facilitate vision-language feature learning. In the architecture design, a particular detail is we don't use actions in the segment for regret learning. The reason is, as shown in previous works (Du et al., 2023; Ajay et al., 2023), the actions can be easily reconstructed by training an inverse-dynamic model from state sequence. As a result, the representation of videos implicitly contains the information of actions.

**Lemma A.1.** *Given a preference dataset $\mathcal{D}_\succ$ that contains all possible $H$-length segments of MDP $\mathcal{M}_r$, let $\Sigma_r^*$ be the set of maximum segments in $\mathcal{D}_\succ$, then the optimal policy $\pi^*(a|s)$ is recovered by segments $\sigma^* \in \Sigma_r^*$ that contain the corresponding state-action pairs.*

*Proof.* For the definition of reward function $r$ in given MDP in $M_r$, let $\sigma^*$ be an optimal segment and $\sigma^{\neg*}$ be any sub-optimal segment, then we have

$$\text{regret}(\sigma^*) < \text{regret}(\sigma^{\neg*}), \tag{12}$$

which leads to $P^*_{\text{regret}}(\sigma^* \succ \sigma^{\neg*}|r) > 0.5$. Generally, for any two segments with $\text{regret}(\sigma^1) < \text{regret}(\sigma^2)$, we have $P^*_{\text{regret}}(\sigma^1 \succ \sigma^2|r) > 0.5$; as for $\text{regret}(\sigma^1) = \text{regret}(\sigma^2)$, we have $P^*_{\text{regret}}(\sigma^1 \succ \sigma^2|r) = 0.5$. As a result, the regret-based preference distribution induces a total ordering over segments in $\mathcal{D}_{\succ}$. Because the regret value has a minimum value $\min \text{regret}(\sigma) = 0$, there exists a set of segments that are ranked highest under this ordering, denoted as $\Sigma^*_r$. As we assume $\mathcal{D}_{\succ}$ contains all possible segments of MDP $\mathcal{M}_r$, these segments are those that achieve the minimum regret and are optimal segments.

Since $\Sigma^*_r$ contains all optimal segments, then for a state-action pair $(s, a) \in \Sigma^*_r$, we have $Q^*_r(s, a) = V^*_r(s)$. We define the $\Pi^*_r$ the set of optimal policies in $\mathcal{M}_r$, then $\Sigma^*_r$ determines the set of state-action pairs such that $Q^*_r(s, a) = V^*_r(s)$, which determines $\Pi^*_r$. As a result, $\Sigma^*_r$ determines $\Pi^*_r$, and the optimal policy can be recovered through a regression on $\Sigma^*_r$. $\square$

In practice, acquiring $\Sigma^*_r$ with the minimum regret value is difficult as it requires a total ordering over segments. As a result, we adopt $P^*_{\text{regret}}$ as a preference distribution to provide (possibly) infinite preference data $\mathcal{D}_r$ and adopt an off-the-shelf preference-based RL algorithm for policy optimization. Theoretically, RL-based reward optimization and contrastive-based preference optimization both aim to minimize regret with respect to the optimal policy (Wang et al., 2023; Cai et al., 2020). Our method that approximates $P^*_{\text{regret}}$ can generate preference data for off-the-shelf preference-based RL by leveraging the generalization ability of vision-language models.

Comparing $P^*_{\text{regret}}$ and $P_\psi$, we also remark that the negative regret $-\text{regret}(\sigma|r) \leq 0$ always holds, while $f_\psi$ that calculates the preference score via a neural network can be either positive or negative. Such an issue can be solved by subtracting the score that the optimal segment (i.e., denoted by $v^*$ for video) obtains in the preference distribution, as

$$\begin{aligned}
\tilde{P}_\psi[v_1 \succ v_2|l] &= \frac{\exp\big(f_\psi(v_1|l) - f_\psi(v^*|l)\big)}{\exp\big(f_\psi(v_1|l) - f_\psi(v^*|l)\big) + \exp\big(f_\psi(v_2|l) - f_\psi(v^*|l)\big)} \\
&= \text{logistic}\big([f_\psi(v_1|l) - f_\psi(v^*|l)] - [f_\psi(v_2|l) - f_\psi(v^*|l)]\big) \\
&= \text{logistic}\big([f_\psi(v_1|l) - f_\psi(v_2|l)]\big) \\
&= P_\psi[v_1 \succ v_2|l],
\end{aligned} \tag{13}$$

where $\tilde{P}_\psi[v_1 \succ v_2|l]$ recover the regret preference distribution, which induces the same preference label as $P_\psi[v_1 \succ v_2|l]$ for the same paired videos. $\text{logistic}(x) = 1/(1 + \exp(-x))$ in Eq. (13).

**The difference between previous preference-based RL methods and VLP from theoretical perspective.** The main theoretical difference between previous preference-based RL methods and VLP is, previous methods adopt a reward-based Bradley-Terry (BT) model by maximizing $\log \text{logistic}(\sum_t r(s_t^{\sigma^1}, a_t^{\sigma^1}) - \sum_t r(s_t^{\sigma^1}, a_t^{\sigma^2}))$, while our method adopts a **regret-based BT** model by maximizing $\log \text{logistic}(\sum_t A(s_t^{\sigma^1}, a_t^{\sigma^1}) - \sum_t A(s_t^{\sigma^1}, a_t^{\sigma^2}))$, where we have $\text{regret}(\sigma) = -\sum_t A(s_t^\sigma, a_t^\sigma)$. In our method, since we construct a dataset with video-level preference, it is more natural to adopt a regret-based BT model that learns $f_\psi(\sigma)$ to approximate the video regret $-\text{regret}(\sigma)$ compared to the optimal trajectory (Proposition 3.1). As a result, we do not explicitly estimate step-wise reward/advantage but directly estimate the overall regret of the whole video. In our theoretical analysis, we prove that such regret-based and video-level preferences also leads to an **optimal policy** under mild conditions (Lemma A.1). To ensure the rigor of the theory, we assume that the accessibility of reward function since both regret and advantage are mathematically defined by the true reward function. In practice, such an assumption is relaxed by incorporating some pseudo-labels that leverage the implicit preference contained in video and vision-language correspondence.

# B  DETAILS OF MTVLP COLLECTION

For the 50 robotic manipulation tasks in Meta-World (Yu et al., 2020), we divide *Button Press*, *Door Close*, *Drawer Close*, *Faucet Close*, and *Window Open* as test tasks and the other 45 tasks as train tasks.

For each task, we leverage scripted policies of Meta-World (Yu et al., 2020) to collect trajectories. For expert trajectories, we add Gaussian noise sampled from $\mathcal{N}(0, 0.1)$. For medium trajectories, we utilize the `near_object` flag returned by each task to determine whether the first subtask is completed and add Gaussian noise sampled from $\mathcal{N}(0, 0.5)$. For random trajectories, the actions are sampled from uniform distribution $\mathcal{U}[0, 1]$. We collect 32 trajectories of each type of trajectory for each task, resulting in a total of 4800 trajectories for all tasks. We query GPT-4V (OpenAI, 2023) to generate language instructions by the prompt containing an example of generating diverse language instructions, an example of generating synonym nouns, task name, task instruction, and an image rendering the task. The detailed prompt we used is shown in Table 9.

Table 9: Prompt for generating diverse language instructions. The verb structures list and synonym nouns example are from Table 2 and Table 4 in LAMP (Adeniji et al., 2023), respectively.

> `System Message`: Suppose you are an advanced visual assistant. Your task is to generate more instructions with the same meaning but different expressions based on the task instruction I provide, generating 40 new instructions for each task. The instructions you generate need to be as simple and clear as possible. Below is an example of an answer for picking up an object. The answer should be formatted as a Python list.
> – Begin of instruction example –
> `Task instruction`: "Pick up the [NOUN]"
> `Answer`:
> `Verb Structures List`
> – End of instruction example –
> Moreover, you need to be mindful to replace the nouns in the instructions with synonyms, such as replacing "bag" with the following words in the Python list:
> – Begin of synonym example –
> `Synonym Nouns`
> – End of synonym example –
> The tasks are from Meta-World benchmark and the image of the task is rendered in a 3D simulation environment. In the environment, there is a wooden table and a robotic arm. The robotic arm is placed above the table. The robotic arm needs to manipulate the object(s) on the table to complete tasks.
> My instruction for `Task Name` task: `Task Instruction`
> `Answer`:

## C EXPERIMENTAL DETAILS

### C.1 TASKS

**Meta-World.** The tasks used in the experiments are from the test tasks of MTVLP. Figure 4 shows these tasks and the task descriptions are as follows:

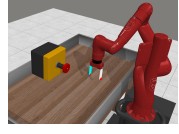 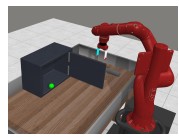 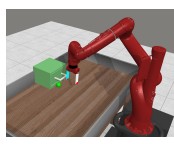 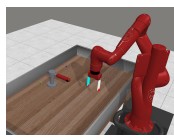 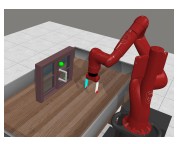

(a) Button Press    (b) Door Close    (c) Drawer Close    (d) Faucet Close    (e) Window Open

Figure 4: Five simulated robotic manipulation tasks used for experimental evaluation.

- Button Press: The goal of the robotic arm is to press the button. The initial position of the arm is randomly sampled.
- Door Close: The goal of the robotic arm is to close the door. The initial position of the arm is randomly sampled.
- Drawer Close: The goal of the robotic arm is to close the drawer. The initial position of the arm is randomly sampled.

- Faucet Close: The goal of the robotic arm is to close the faucet. The initial position of the arm is randomly sampled.

- Window Open: The goal of the robotic arm is to open the window. The initial position of the arm is randomly sampled.

## C.2 IMPLEMENTATION DETAILS

We implement our method based on the publicly released repository of LAPP (Xie et al., 2023).[1] Following LAPP (Xie et al., 2023), we use a pre-trained ViT-B/16 CLIP (Radford et al., 2021) model as our video encoder and language encoder. To achieve efficient learning, we uniformly sample 8 frames to represent each video. The detailed hyperparameters of our method are shown in Table 10. Training a VLP model takes about 6 hours on a single NVIDIA RTX 4090 GPU with 12 CPU cores and 120 GB memory, without costly pre-training process like VLM reward or VLM preference methods (Nair et al., 2023; Ma et al., 2023b;a).

Table 10: Hyperparameters of VLP.

| Hyperparameter | Value |
|---|---|
| Prediction head | $(512, 256)$ |
| Number of self-attention layers | 2 |
| Number of attention heads | 16 |
| Batch size | 16 |
| Optimizer | Adam |
| Learning rate | 3e-5 |
| Learning rate decay | cosine decay |
| Weight decay | 0.1 |
| Dropout | 0.1 |
| Number of epochs | 15k |
| Number of negative samples | 4 |
| Number of video frames | 8 |
| Weight of ILP loss $\lambda_1$ | 0.1 |
| Weight of IVP loss $\lambda_1$ | 0.5 |

IQL, P-IQL, IPL and CPL are implemented based on the official repository of CPL and IPL.[2][3] The hyperparameters of offline RL and RLHF algorithms are listed in Table 11, Table 12, and Table 13. For the inference of VLP labels, we first use K-means clustering to divide the trajectories of each test task into 2 sets, following Liu et al. (2024b). Then we sample 100 trajectory segments of length 50 from each set to construct segment pairs and predict preference labels of these pairs with trained VLP model. Training RL and RLHF algorithms take about 10 minutes using a single NVIDIA RTX 4090 GPU with 6 CPU cores and 60 GB memory.

For VLM methods, R3M, VIP, LIV, and VLM-RM are implemented based on their official repositories.[4][5][6][7] The CLIP baseline is a variant of VLM-RM and is implemented based on the code of VLM-RM. The language inputs of the VLM baselines except are as listed in Table 14. R3M, LIV, CLIP, and RoboCLIP only require the target column as language inputs, while VLM-RM additionally needs a baseline as a regularization term. R3M requires an initial image and we use the first frame of each trajectory as the initial image, while VIP requires a goal image for VLM rewards inference and we use the last frame of expert videos.

---

[1]https://github.com/amberxie88/lapp
[2]https://github.com/jhejna/cpl
[3]https://github.com/jhejna/inverse-preference-learning
[4]https://github.com/facebookresearch/r3m
[5]https://github.com/facebookresearch/vip
[6]https://github.com/penn-pal-lab/LIV
[7]https://github.com/AlignmentResearch/vlmrm

Table 11: Shared hyperparameters.

| Hyperparameter | Value |
|---|---|
| Network architecture | $(256, 256)$ |
| Optimizer | Adam |
| Learning rate | 1e-4 (CPL), 3e-4 (IQL, IPL and P-IQL) |
| Batch size | 64 |
| Discount | 0.99 |
| Dropout | 0.25 |
| Training steps | 100000 |
| Segment length | 50 (RLHF) |
| Number of queries | 100 (RLHF) |
| Temperature | 0.3333 (IQL, IPL and P-IQL) |
| Expectile | 0.7 (IQL, IPL and P-IQL) |
| Soft target update rate | 0.005 (IQL, IPL and P-IQL) |

Table 12: Hyperparameters of CPL.

| Hyperparameter | Value |
|---|---|
| Temperature | 0.1 |
| Contrastive bias | 0.5 |
| BC weight | 0.0 |
| BC steps | 10000 |

Table 13: Hyperparameters of IPL (left) and P-IQL (right).

| Hyperparameter | Value | Hyperparameter | Value |
|---|---|---|---|
| Regularization weight | 0.5 | Reward learning steps | 30 |

Table 14: Language inputs used for evaluating VLM baselines on the test tasks.

| Task | Target | Baseline (Only for VLM-RM) |
|---|---|---|
| Button Press | press button | button |
| Door Close | close door | door |
| Drawer Close | close drawer | drawer |
| Faucet Close | turn faucet left | faucet |
| Window Open | move window left | window |

# D    ADDITIONAL EXPERIMENTS

**Preference Label Accuracy.**    To compare the relative relation of VLM rewards with VLP, we compute the preference label accuracy of each method. The accuracy is measured by comparing the predicted preference labels with scripted preference labels. The results in Table 15 show that VLP exceeds the VLM baselines by a large margin, demonstrating VLM rewards do not capture the relative reward relationship.

Table 15: Preference label accuracy of VLP against VLM baselines. The accuracy of our method is shaded and the best score of all methods is **bolded**.

| Task | R3M | VIP | LIV | CLIP | VLM-RM (0.0) | VLM-RM (1.0) | RoboCLIP | VLP |
|---|---|---|---|---|---|---|---|---|
| Button Press | 91.0 | 40.0 | 62.0 | 53.0 | 62.0 | 41.0 | 46.0 | **93.0** |
| Door Close | 98.0 | 57.0 | 97.0 | 49.0 | 59.0 | 10.0 | 61.0 | **100.0** |
| Drawer Close | 66.0 | 49.0 | 39.0 | 66.0 | 65.0 | 58.0 | 43.0 | **96.0** |
| Faucet Close | 98.0 | **100.0** | 97.0 | 38.0 | 25.0 | 65.0 | 63.0 | **100.0** |
| Window Open | 72.0 | 88.0 | 16.0 | 81.0 | 88.0 | 16.0 | 49.0 | **98.0** |
| **Average** | 85.0 | 66.8 | 62.2 | 57.4 | 59.8 | 38.0 | 52.4 | **97.4** |

**Different VLMs/LLMs for Language Instruction Generation.**    To see the influence of different language model on our method, we we conduct additional experiments using instructions from less capable model, such as GPT-3.5 and open-source Llama-3.1-8B-Instruct. We observe that generating diverse language instructions does not necessarily require strong VLMs like GPT-4V, even open-source Llama-3.1-8B-Instruct can accomplish this job since the language model is prompted with a diverse set of examples, following LAMP (Adeniji et al., 2023). The results in Table 16 show that the model's performance is relatively stable across different LLMs.

Table 16: Preference label accuracy of VLP with language instructions generated by different VLMs/LLMs.

| Task | GPT-4V | GPT-3.5 | Llama-3.1-8B-Instruct |
|---|---|---|---|
| Button Press | 93.0 | 93.0 | 91.0 |
| Door Close | 100.0 | 100.0 | 98.0 |
| Drawer Close | 96.0 | 96.0 | 97.0 |
| Faucet Close | 100.0 | 100.0 | 100.0 |
| Window Open | 98.0 | 99.0 | 99.0 |
| **Average** | 97.4 | 97.6 | 97.0 |

# E    DISCUSSIONS

**How do ILP and IVP benefit VLP?**    The inclusion of ILP and IVP in our training data serves critical roles in enhancing the generalization and robustness of our model. ILP allows our model to learn to disregard language variations when they do not impact the preference outcomes, thus training the model to focus on task-relevant features rather than linguistic discrepancies. On the other hand, IVP facilitates the model's ability to generalize across different tasks by learning to associate videos with their corresponding task-specific language instructions effectively. This capability is crucial when the model encounters new tasks or language contexts, as it must discern relevant from irrelevant information to make accurate preference predictions. By training with both ILP and IVP, our model learns a more holistic understanding of the task space, which not only improves its performance on seen tasks but also enhances its adaptability to new, unseen tasks or variations in task descriptions, as evidenced by our experimental results where the model demonstrated generalization capabilities.

**How does different train-test split influence VLP?**    We conduct experiments on the Meta-World ML45 benchmark, training the vision-language preference model on its training tasks and evaluating on its test tasks. We compute VLP label accuracy by comparing VLP label with scripted preference labels. The results shown in Table 17 demonstrate the strong generalization capability of our method

on unseen tasks in ML45. This reinforces the robustness and adaptability of our framework regardless of task split.

Table 17: Preference label accuracy of VLP on ML45 test tasks.

| Task | VLP Acc. |
|------|----------|
| Bin Picking | 95.0 |
| Box Close | 90.0 |
| Door Lock | 100.0 |
| Door Unlock | 100.0 |
| Hand Insert | 100.0 |
| **Average** | 97.0 |

