# OpenReview forum: "VLP: Vision-Language Preference Learning for Embodied Manipulation"
_ICLR.cc/2025/Conference — Submitted to ICLR 2025_

### Official Review · Reviewer_R4Zt · 2024-11-01

**Soundness:** 3
**Presentation:** 4
**Contribution:** 3
**Rating:** 5
**Confidence:** 3

**Summary:**

This paper introduces VLP, a vision-language preference learning framework, where the preference model is designed to generalize across unseen tasks. The preference model is based on an open-sourced CLIP model augmented with a trainable component. The model is trained using different types of preferences, categorized as ITP, ILP, and IVP. The experiments on the Meta-World benchmark support the paper's claims.

**Strengths:**

- The division of video-language preference types is novel and thoughtfully structured. ITP reflects traditional preferences, while IVP appears to enhance the model’s instruction-following capabilities. ILP seems to serve as a regularizer, adding robustness to the model. This categorization is well-conceived.
- The writing is clear, and the graphical illustrations effectively convey the content, enhancing overall readability.
- Both theoretical analysis and empirical findings support the framework.

**Weaknesses:**

- A primary concern is the rationale behind the train-test task split in Meta-World. While the experimental results favor the proposed framework, it is unclear if the task split was specifically selected for favorable outcomes. Using Meta-World's ML45 benchmark, which provides a pre-defined split for comparability across works, could enhance the reproducibility and rigor of the results. Clarifying this point would strengthen the paper, and I would be inclined to raise my score if this concern is addressed, as the rest of the experimental design is robust.

**Questions:**

- The authors claim novelty in the architecture (line 071), yet it is not immediately clear what sets it apart, and this assertion seems somewhat overstated. Could the authors clarify the specific architectural innovations that distinguish this approach?

---

> ### Author Response · Authors · 2024-11-23
> **Rebuttal**
>
> We thank reviewer R4Zt for the constructive comments. We will give our point-wise responses below.
>
> **W1: "A primary concern is the rationale behind the train-test task split in Meta-World ... as the rest of the experimental design is robust."**
>
> > **A**: Thank you for your valuable feedback. The test tasks used in our paper are motivated by the test tasks in prior preference-based RL works [1,2,3,4,5]. However, we agree that using the standard ML45 benchmark can provide additional rigor and ensure broader reproducibility. To address this, we conduct experiments on the ML45 benchmark, training the vision-language preference model on its training tasks and evaluating on its test tasks. The results shown below demonstrate the strong generalization capability of our method on unseen tasks in ML45. This reinforces the robustness and adaptability of our framework regardless of task split. We have updated the manuscript and included these results and discussions in Appendix E.
> >
> > | Task | VLP Accuracy |
> > | - | - |
> > | Bin Picking | 95.0 |
> > | Box Close | 90.0 |
> > | Door Lock | 100.0 |
> > | Door Unlock | 100.0 |
> > | Hand Insert | 100.0 |
> > | Average | 97.0 |
>
> **Q1: "The authors claim novelty in the architecture (line 071) ... clarify the specific architectural innovations that distinguish this approach?"**
>
> > **A**: Thank you for highlighting this point. The architectural novelty lies in three key aspects that distinguish our approach from prior methods:
> >
> > 1) Vision-Language Preference Definition: We introduce language-conditioned preferences (intra-task, inter-language, and inter-video preferences) that generalizes the preference learning framework. Unlike prior work focused on state-based preferences [1,2,3,4,5], our method leverages video trajectories and language as flexible and universal conditioning inputs to define and learn preferences from multi-modal data.
> > 2) The Preference Alignment Objective:
> > The training objective explicitly optimizes the model to align video and language features according to our defined preference types. This alignment ensures that the learned preferences generalize effectively across tasks and instructions.
> > 3) Cross-Modal Attention:
> > The proposed vision-language preference model integrates video and language modalities using a cross-modal attention mechanism rather than naively combining video and language features. This mechanism not only extracts relevant features from both modalities but also aligns them to predict trajectory-wise preferences.
> >
> > We have revised the manuscript to clearly highlight these points and their impact in Appendix E of the revised draft.
>
> **References**
>
> [1] PEBBLE: Feedback-Efficient Interactive Reinforcement Learning via Relabeling Experience and Unsupervised Pre-training. ICML 2021.
>
> [2] SURF: Semi-supervised Reward Learning with Data Augmentation for Feedback-efficient Preference-based Reinforcement Learning. ICLR 2022.
>
> [3] Reward Uncertainty for Exploration in Preference-based Reinforcement Learning. ICLR 2022.
>
> [4] Few-Shot Preference Learning for Human-in-the-Loop RL. CoRL 2023.
>
> [5] PEARL: Zero-shot Cross-task Preference Alignment and Robust Reward Learning for Robotic Manipulation. ICML 2024.

---

> > ### Comment · Reviewer_R4Zt · 2024-11-26
> >
> > I appreciate the significant effort the authors have put into addressing the concerns raised during the review process.
> >
> > - While I understand the motivation behind the references cited, I believe they are not entirely appropriate for comparison, as they primarily address single-task setups. For instance, [1] involves training and test on individual tasks with randomized agent resets and goal positions. In contrast, this paper focuses on training the preference model on training tasks and evaluating its generalizability on test tasks. In this meta-learning context, selecting a specific train-test task split is not a suitable evaluation criterion. I acknowledge and appreciate the additional experiments conducted during the rebuttal period; however, they do not sufficiently address this concern, as they lack the comprehensive comparison analysis needed to compensate for the limitations of the original experimental setup.
> > - I acknowledge the preference definition and objective is novel and original. However, I am still not convinced that the architecture of neural network is novel. Cross-modal attention is not a novel concept to make the architecture novel. As the novelty is sufficient from the preference definition and objective, I believe it is better not to claim architectural novelty.
> >
> > As my primary concern regarding the rationale for the train-test split remains unresolved, I believe this paper is not yet ready for publication. Regrettably, I am unable to raise my score at this time.

---

> > > ### Author Response · Authors · 2024-11-28
> > > **Further responsse to Reviewer R4Zt**
> > >
> > > Thank you for your insightful feedback. We understand your concern regarding the train-test split in the Meta-World environment and the novelty of cross-modal attention. We would like to clarify the following points:
> > >
> > > 1. We agree that the prior works we referenced mainly focus on single-task setups, which are different from the multi-task evaluation in our paper. However, CriticGPT [1], which is another preference-based RL method evaluated in similar multi-task setting, also uses the Meta-World benchmark and manually selects its own train-test task split. We believe that there is no universally accepted "gold standard" for how to split tasks in multi-task setting and the train-test split is just a design choice instead of a standardized procedure. Additionally, the baselines are evaluated using the same split for fair comparison.
> > > 2. Regarding the novelty of cross-modal attention, we agree that attention mechanism itself is not new. However, the key novelty of our work lies in the vision-language preference model we introduce. Unlike previous preference-based RL methods, which typically rely on state representations as vectors, our model learns preferences directly from the alignment of visual and language features. This approach is distinct and contributes to the overall advancement of preference-based RL. We will revise the manuscript to make this distinction clearer.
> > >
> > > We hope these clarifications help address your concerns. If you have any further questions, we would be happy to discuss them.
> > >
> > > **References**
> > >
> > > [1] Enhancing Robotic Manipulation with AI Feedback from Multimodal Large Language Models.

---

> > > ### Author Response · Authors · 2024-12-03
> > > **A gentle reminder**
> > >
> > > Dear Reviewer R4Zt,
> > >
> > > As the deadline of the discussion period draws near, we would greatly appreciate your attention to our rebuttal. We would like to know whether we have adequately addressed your concerns. Your feedback is crucial to us, and we value the opportunity to address any concerns you have raised. If you have any further questions, we are more than happy to discuss.
> > >
> > > Thank you for your time and consideration.
> > >
> > > Best regards,
> > >
> > > Authors

---

### Official Review · Reviewer_NSgJ · 2024-11-02

**Soundness:** 3
**Presentation:** 4
**Contribution:** 3
**Rating:** 6
**Confidence:** 4

**Summary:**

This paper introduces VLP (Vision-Language Preference learning), a framework for learning general preference feedback for embodied manipulation tasks. The key contribution is a vision-language preference model that provides feedback by aligning video and language modalities. The paper presents extensive empirical evaluation, demonstrating strong performance and generalization capabilities to unseen tasks and language instructions.

**Strengths:**

1. The paper presents an effective framework that combines vision-language alignment with preference learning for robotic manipulation tasks. The experimental results show consistent improvements over VLM-based approaches across multiple tasks and demonstrate good generalization performance.
2. The paper is well-structured and easy to follow, presenting its ideas clearly.

**Weaknesses:**

1. The evaluation is limited to relatively simple Meta-World tasks, without testing on more complex task domains (e.g., MANISKILL2 [1] and MyoSuite [2]).
2. The paper lacks comparison with human preference labels, which would validate the quality of the generated preferences against human intent.
3. The theoretical analysis assumes access to all possible segments, weakening its practical implications.
4. (minor) The paper does not report the performance of scripted policies, which would help establish an upper bound for task performance and validate the quality of collected expert demonstrations.

[1] Gu, Jiayuan, et al. "Maniskill2: A unified benchmark for generalizable manipulation skills." *arXiv preprint arXiv:2302.04659* (2023).

[2] Caggiano, Vittorio, et al. "MyoSuite--A contact-rich simulation suite for musculoskeletal motor control." *arXiv preprint arXiv:2205.13600* (2022).

**Questions:**

1. How does the computational cost of training VLP compare to other approaches like R3M or VIP?
2. How sensitive is the model to the quality and diversity of language instructions? Is there a significant performance drop when using instructions generated by a less capable model than GPT-4V?

---

> ### Author Response · Authors · 2024-11-23
> **Rebuttal (Part 1/2)**
>
> We thank reviewer NSgJ for the positive support and constructive comments. We will give our point-wise responses below.
>
> **W1: "The evaluation is limited to relatively simple Meta-World tasks, without testing on more complex task domains (e.g., MANISKILL2 [1] and MyoSuite [2])."**
>
> > **A**: Please refer to the subsequent glogal response.
>
> **W2: "The paper lacks comparison with human preference labels, which would validate the quality of the generated preferences against human intent."**
>
> > **A**: We appreciate this suggestion. Although previous preference-based RL works mainly evaluate their methods using scripted preference labels, we agree that comparing our learned preferences with human labels would provide additional insights into the quality of the generated preferences. To address this, we collect human preference labels and conduct experiments with CPL, IPL and PIQL. The results in the following table show that our model's preferences align well with human intent, achieving high accuracy across different tasks. We have included these results in the Table 2 of the revised version to demonstrate the effectiveness of our method in learning preferences.
> >
> > | **Task** | **P-IQL Human** | **P-IQL Scripted** | **P-IQL VLP** | **IPL Human** | **IPL Scripted** | **IPL VLP** | **CPL Human** | **CPL Scripted** | **CPL VLP** | **VLP Acc. Human** | **VLP Acc. Scripted** |
> > |-|-|-|-|-|-|-|-|-|-|-|-|
> > | Button Press | 93.1 ± 5.2 | 72.6 ± 7.1 | 90.1 ± 3.9 | 65.2 ± 7.2 | 50.6 ± 7.9 | 56.0 ± 1.4 | 85.0 ± 7.2 | 74.5 ± 8.2 | 83.9 ± 11.8 | 99.0  | 93.0  |
> > | Door Close   | 79.2 ± 6.3 | 79.2 ± 6.3 | 79.2 ± 6.3 | 61.5 ± 9.4 | 61.5 ± 9.4 | 61.5 ± 9.4 | 98.5 ± 1.0 | 98.5 ± 1.0 | 98.5 ± 1.0  | 100.0 | 100.0 |
> > | Drawer Close | 63.7 ± 6.4 | 49.3 ± 4.2 | 64.9 ± 2.9 | 63.2 ± 4.6 | 64.3 ± 9.6 | 63.2 ± 4.7 | 54.1 ± 8.7 | 45.6 ± 3.5 | 57.5 ± 14.3 | 96.0  | 96.0  |
> > | Faucet Close | 51.1 ± 7.5 | 51.1 ± 7.5 | 51.1 ± 7.5 | 45.4 ± 8.6 | 45.4 ± 8.6 | 45.4 ± 8.6 | 80.0 ± 2.9 | 80.0 ± 2.9 | 80.0 ± 2.9  | 100.0 | 100.0 |
> > | Window Open  | 69.7 ± 6.8 | 62.4 ± 6.4 | 69.7 ± 6.8 | 61.4 ± 8.6 | 54.1 ± 6.7 | 61.4 ± 8.6 | 99.1 ± 1.1 | 91.6 ± 1.7 | 99.1 ± 1.1  | 100.0 | 98.0  |
> > | **Average**  | 71.4       | 62.9       | 71.0       | 59.3       | 55.2       | 57.5       | 83.3       | 78.0       | 83.8        | 99.0  | 97.4  |
>
> **W3: "The theoretical analysis assumes access to all possible segments, weakening its practical implications."**
>
> > **A**: We acknowledge that the theoretical analysis requires the assumption of access to all possible trajectory segments, which introduces a gap between the theoretical guarantees and empirical applications. However, this assumption is essential for deriving the theoretical results. To address this, we presented the main theoretical analysis in the appendix, and we focus on the empirical results in the main paper to demonstrate the practical utility of the method. To solve your concern, we have clarified this point in the revised draft in the limitations in Section 6.
>
> **W4: "(minor) The paper does not report the performance of scripted policies, which would help establish an upper bound for task performance and validate the quality of collected expert demonstrations."**
>
> > **A**: Thank you for highlighting this. We have conducted experiments using scripted policies on the same set of tasks used in the main experiments. The results in the following table show that the scripted policies achieve $100\%$ success rate on all tasks, providing an upper bound for task performance. Also, we have included these results in the revised version of the draft.
> >
> > | Task | VLP Accuracy |
> > | - | - |
> > | Button Press | 100.0 |
> > | Door Close | 100.0 |
> > | Drawer Close | 100.0 |
> > | Faucet Close | 100.0 |
> > | Window Open | 100.0 |
> > | Average | 100.0 |

---

> ### Author Response · Authors · 2024-11-23
> **Rebuttal (Part 2/2)**
>
> **Q1: "How does the computational cost of training VLP compare to other approaches like R3M or VIP?"**
>
> > **A**: Currently, we train VLP on Meta-World tasks using a single NVIDIA RTX 4090 GPU with 12 CPU cores for approximately 6 hours. For VLM methods like R3M and VIP, they usually requires pre-training on large-scale datasets like Ego4D, which can take several days with the same hardwares. However, since there lacks a large-scale preference dataset for embodied manipulation tasks, we cannot provide a direct comparison of the training cost between VLP and R3M or VIP.
>
> **Q2: "How sensitive is the model to the quality and diversity of language instructions? Is there a significant performance drop when using instructions generated by a less capable model than GPT-4V?"**
>
> > **A**: Thanks for the question. We observe that generating diverse language instructions does not necessarily require strong VLMs like GPT-4V, even open-source Llama-3.1-8B-Instruct can accomplish this job since the language model is prompted with a diverse set of examples, following LAMP [1]. To evaluate this sensitivity, we conduct additional experiments using instructions from less capable model, such as GPT-3.5 and open-source Llama-3.1-8B-Instruct. The results in the following table show that the model's performance is relatively stable across different LLMs. We have included these results in the revised version in Appendix D to demonstrate the model's robustness to different types of models for generating language instructions.
> >
> > | Task | GPT-4V | GPT-3.5 | Llama-3.1-8B-Instruct |
> > | - | - | - | - |
> > | Button Press | 93.0 | 93.0 | 91.0 |
> > | Door Close | 100.0 | 100.0 | 98.0 |
> > | Drawer Close | 96.0 | 96.0 | 97.0 |
> > | Faucet Close | 100.0 | 100.0 | 100.0 |
> > | Window Open | 98.0 | 99.0 | 99.0 |
> > | Average | 97.4 | 97.6 | 97.0 |
>
> **References**
>
> [1] Language Reward Modulation for Pretraining Reinforcement Learning. ArXiv 2023.

---

> ### Author Response · Authors · 2024-12-03
> **A gentle reminder**
>
> Dear Reviewer NSgJ,
>
> As the deadline of the discussion period draws near, we would greatly appreciate your attention to our rebuttal. We would like to know whether we have adequately addressed your concerns. Your feedback is crucial to us, and we value the opportunity to address any concerns you have raised. The answer to **W1** has been provided in the global response. If you have any further questions, we are more than happy to discuss.
>
> Thank you for your time and consideration.
>
> Best regards,
>
> Authors

---

### Official Review · Reviewer_LWSo · 2024-11-04

**Soundness:** 2
**Presentation:** 3
**Contribution:** 3
**Rating:** 5
**Confidence:** 4

**Summary:**

This paper proposes a novel video-based, vision-language-interleaved preference learning method for robotic control, named VLP. It defines three types of language-conditioned preferences: ITP, ILP, and IVP. The authors introduce a novel vision-language preference alignment framework that includes a learnable cross-modal transformer model to fuse video tokens and language tokens. They constructed a vision-language preference dataset with clear intra-task preference relations, MTVLP, containing 4.8K videos. Experimental results demonstrate the superiority of VLP compared to other RLHF methods with scripted labels or other vision-language rewards. Additionally, empirical evidence suggests that ILP and IVP, alongside the traditional ITP, contribute to improved performance, and that the 4.8K videos are both necessary and sufficient to achieve over 97% ITP accuracy.

**Strengths:**

* This paper presents strong empirical evidence and extensive experiments supporting the proposed approach.
* The novel cross-modal architecture effectively fuses video and language through learnable parameters to compute preferences.
* Furthermore, the introduction of language-conditioned preferences, namely Intra-Task Preference (ITP), Inter-Language Preference (ILP), and Inter-Video Preference (IVP), is a notable contribution that enhances the model's adaptability across different scenarios.

**Weaknesses:**

* The theoretical claim seems to lack clear logical reasoning to justify the assertion that "the proposed preference model can be considered as parameterized negative regret that approximates the true negative regret of the whole segment". Although Eq. (10) and Eq. (11) have similar shapes, that does not mean that one approximates the other.
* I'm concerned that the simplicity of ILP and IVP definitions may limit VLP's generalizability. The preference labels defined in Table 1 overlook potential similarities between videos or language instructions across different tasks: They can assign a negative signal even if two videos from different tasks are similar (or in the case where the video and language from different tasks are semantically related) This approach may only work effectively within a carefully selected task distribution, potentially weakening the paper's claims of generalizability.

**Questions:**

* Comparing this work with RoboCLIP (Sontakke et al., 2023) may provide valuable context. Baselines in this paper lack video input, so VLP’s advantage might come from its temporal reasoning. While VLP is compute-efficient, RoboCLIP is zero-shot. Demonstrating the cross-modal architecture’s distinct benefits would strengthen the claims.
* Regarding the second weakness: (1) How are video pairs in MTVLP constructed for ITP, ILP, and IVP regarding optimality levels? Are all combinations (e.g., expert, medium, random) considered? (2) Could you provide more examples of how medium-optimality is defined across the 50 tasks? Do any tasks share similar initial subtasks?
* Writing clarification suggestions: (1) It would be great if Table 1 is accompanied by v_i^j and l^k notations. (2) In several places, “language” is used to mean "language instructions," which might cause confusion. For instance, "unseen language" might imply a different spoken language rather than new instructions in English.

---

> ### Author Response · Authors · 2024-11-23
> **Rebuttal (Part 1/2)**
>
> We thank reviewer LWSo for the constructive comments. We will give our point-wise responses below.
>
> **W1: "The theoretical claim seems to lack ... not mean that one approximates the other."**
>
> > **A**: Thanks for the question. We give further justification for this point in the following.
> >
> > In the theoretical analysis, we define $A^*_r(s_t^\sigma,a_t^\sigma)$ as the optimal advantage function of $\pi^*$ under the reward $r$. Then the regret preference model is defined as
> >
> > $P\_{\rm regret}^*(\sigma^1 \succ \sigma^2 | r) = \frac{\exp \sum\_{\sigma^1} A^*\_r(s^{\sigma^1}\_t,a^{\sigma^1}\_t)}{\exp \sum\_{\sigma^1} A^*\_r(s^{\sigma^1}\_t,a^{\sigma^1}\_t) + \exp \sum\_{\sigma^2} A^*\_r(s^{\sigma^2}\_t,a^{\sigma^2}\_t)}.$
> >
> > Then we denote the regret function as $-{\rm regret}(\sigma)=g\_\varphi(\sigma)=\sum\_\sigma A^*\_r(s^\sigma\_t,a^\sigma\_t)$, where the summation is calculated over the timeteps of segment $\sigma$.
> >
> > In this case, we define the optimization objective as
> >
> > $L\_1(g) = \mathbb{E}\_{\sigma\_1, \sigma\_2 \sim D\_r} {\rm CE} [ P\_{\rm regret}^{*}(\sigma^1 \succ \sigma^2 \mid r) ] = \mathbb{E}\_{\sigma\_1, \sigma\_2 \sim D\_r} {\rm CE} [\frac{\exp (g\_\varphi(\sigma^1))}{\exp (g\_\varphi(\sigma^1)) + \exp (g\_\varphi(\sigma^2))}],$
> >
> > then we obtain that value of $g_\phi(\sigma)$ that estimates the negative regret function, i.e., we can obtain $g\_\varphi^*=\arg\min L\_1(g\_\varphi)$.
> >
> > According to Eq. (11), we rewrite the actual optimization objective of our learning objective with samples from $D\_r$ as
> >
> > $L\_2(f\_\psi) = \mathbb{E}\_{\sigma\_1,\sigma\_2 \sim D\_r} {\rm CE} [\frac{\exp (f\_\psi(\sigma^1))}{\exp (f\_\psi(\sigma^1)) + \exp (f\_\psi(\sigma^2))}].$
> >
> > As a result, if $g_\varphi(\sigma)=f_\psi(\sigma)$, we have $L_1(g)=L_2(f_\psi)$. We also have $f^*_\psi=\arg\min L_2(f_\psi)=\arg\min L_1(g)=g_\varphi^*$ for the same preference dataset $D_r$. As a result, given the same preference dataset $D_r$, the proposed preference model $f_\psi$ **can be considered** as parameterized negative regret that approximates the true negative regret of the segment if both $L_1(g)$ and $L_2(f)$ are minimized.
> >
> > However, in practice, $f_\psi(\sigma)$ and $g_\varphi(\sigma)$ are optimized under difference preference dataset. Specifically, $g_\varphi$ is defined under the preference dataset labeled by the ground truth reward function. In contrast, $f_\psi$ is optimized with preference data that has pseudo-labels $\tilde{y} \in \\{y^{\text{ITP}}, y^{\text{ILP}}, y^{\text{IVP}}\\}$ according to the optimality of segment and segment-language correspondence, which is an approximation of the ground truth preference labels.
>
> **W2: "I'm concerned that the simplicity ... weakening the paper's claims of generalizability."**
>
> > **A**: We appreciate it for pointing out this concern. However, our framework explicitly focuses on learning **relative** preference relationships rather than absolute semantic similarities.
> >
> > 1) The preference model prioritizes alignment between videos and language instructions that are more relevant to each other. Even if a video and a language instruction from different tasks have some degree of similarity, our language-conditioned preferences do not treat such cases as positive examples compared with more similar pairs.
> >
> > 2) While ILP and IVP definitions rely on language conditioning, the cross-modal transformer actively aligns semantic relations between videos and language. This enables the model to capture and exploit similarities even when tasks differ, leveraging the flexibility of language-conditioned preferences.
> >
> > 3) Empirical results show that VLP effectively handles unseen tasks and languages instructions (as reported in Section 5). This demonstrates that our framework successfully learns generalized preferences rather than rigid task-specific labels.

---

> ### Author Response · Authors · 2024-11-23
> **Rebuttal (Part 2/2)**
>
> **Q1: "Comparing this work with RoboCLIP ... would strengthen the claims."**
>
> > **A**: Thank you for the suggestion. We further conduct experiments using RoboCLIP on the five evaluation tasks and the results are shown in Table 4 of the updated draft. The comparison demonstrates that VLP outperforms RoboCLIP, highlighting the benefits of our vision-language preference learning. We have included a detailed comparison with RoboCLIP in the revised version to provide a more comprehensive analysis of the advantages of VLP over zero-shot video-input models.
>
> **Q2: "Regarding the second weakness ... share similar initial subtasks?"**
>
> > **A**: Thank you for the questions.
> >
> > (1) In the construction of video pairs for ITP, ILP, and IVP in the MTVLP dataset, we consider all combinations of video optimality levels: expert, medium, and random. Specifically:
> >
> > - For ITP, we pair videos from the same task, where we assign preference based on the optimality of the trajectories (i.e., expert > medium > random).
> > - For ILP, videos from the same task are paired with language instructions from different tasks. These video pairs are treated as equally preferred under the given language instruction.
> > - For IVP, we pair videos from different tasks with a language instruction from either task, where the preference is assigned based on the alignment of video and task.
> >
> > (2) Regarding medium-optimality, we define medium-optimality as a trajectory that completes part of the task (but not all). For example, in the task of Drawer Open, a medium level trajectory involves grasping the drawer handle but not fully pulling it out. In the task of Hammer, a medium level trajectory completes grasping the hammer but not hitting the nail. For initial subtasks, the task of Drawer Open and Door Open share similar initial subtasks, where the medium level trajectory for Drawer Open involves grasping the drawer handle but not fully pulling it out, while the medium level trajectory for Door Open involves grasping the door handle but not fully opening the door. However, although the initial subtasks may be similar, they have other distinct aspects that the target object is different during the initial subtask. We will provide more detailed examples of medium-optimality across the 50 tasks in the revised version to clarify this definition.
>
> **Q3: "Writing clarification suggestions."**
>
> > **A**: Thanks for these suggestions. For (1), we have updated the manuscript to add the notations in Table 1, which is more clearer than describing with words. For (2), we have revised the confusious "language" term into "language instruction(s)" throughout the paper.

---

> ### Author Response · Authors · 2024-12-03
> **A gentle reminder**
>
> Dear Reviewer LWSo,
>
> As the deadline of the discussion period draws near, we would greatly appreciate your attention to our rebuttal. We would like to know whether we have adequately addressed your concerns. Your feedback is crucial to us, and we value the opportunity to address any concerns you have raised. If you have any further questions, we are more than happy to discuss.
>
> Thank you for your time and consideration.
>
> Best regards,
>
> Authors

---

> ### Comment · Reviewer_LWSo · 2024-12-03
> **Response with Additional Questions**
>
> Thank you for the clarification and revisions. I was impressed with the comparisons with RoboCLIP. Also, regarding Q2, I see your point about the target object being different for each task. The Meta-World tasks appear to have different preconditions (e.g., a door is closed for the "Door Open" task) that can be easily verified using a video, making it difficult to find similar video segments from two different tasks. I agree that providing detailed examples of medium-optimality across the 50 tasks will greatly help clarify this issue.
>
> Nevertheless, I would like to raise some additional questions regarding the theoretical analysis:
>
> **W1-continued:**
>
> I believe that the preference dataset used to optimize the preference model is crucial for determining the meaning of the learned preference model. Can we say that $f_{\psi}$ can be considered as $g_{\varphi}$, which approximates the true negative regret, even if it is optimized with a different preference dataset?
>
> To illustrate my point, here is an example with a similar argument, using the same preference dataset for optimization.
> Let's think of an "immediate reward function", defined by $h(\sigma):= r(s_1^{\sigma}, a_1^{\sigma}) $ for $\sigma = (s_1^{\sigma}, a_1^{\sigma}, \cdots, s_H^{\sigma}, a_H^{\sigma})$. Then we can derive a preference distribution using $h$.
> $$
> P_{\text {imm.rew}} \left(\sigma^1 \succ \sigma^2 \mid r\right)=
> \frac{\exp \left(h \left(\sigma^1\right)\right)}{\exp \left(h \left(\sigma^1\right)\right)+\exp \left(h \left(\sigma^2\right)\right)}
> $$
>
> Similarly to $L_1$ and $L_2$, we can define the optimization objective for $h_{\phi}$ as follows.
>
> $$
> L_3(h_{\phi})=\mathbb{E}\_{\sigma_1, \sigma_2 \sim D_r} \mathrm{CE}\left[
> P\_{\text {imm.rew}}\left(\sigma^1 \succ \sigma^2 \mid r\right)\right]=\mathbb{E}\_{\sigma_1, \sigma_2 \sim D_r} \mathrm{CE}\left[\frac{\exp \left(h_{\phi}\left(\sigma^1\right)\right)}{\exp \left(h_{\phi}\left(\sigma^1\right)\right)+\exp \left(h_{\phi}\left(\sigma^2\right)\right)}
> \right]
> $$
>
> If we use the same preference dataset $D_r$ to optimize $h_\phi$, we will get $h_\phi^*=f_\psi^*=g_{\varphi}^*$. But this result, suggesting that the parameterized immediate reward is equivalent to the parameterized negative regret, sounds weird. I believe the problem arises from the assumption that the same preference dataset can be used for optimization. And if we use different preference datasets for optimization, we can no longer say that each parameterized preference model approximates the other.
>
> Please kindly correct me if I have misunderstood or gotten something wrong.
>
> **Q4:**
> Additionally, it would be helpful to discuss the ground-truth reward function associated with the preference model $f_{\psi}$ to analyze the meaning of the learned preference distribution $P_{\psi}$.
>
> In my understanding, $f_{\psi}$ is defined as a multi-task preference model that can compare two segments from either the same task or different tasks (e.g., $\sigma_1^1 \sim \mathcal{T}^1, \sigma_1^2 \sim \mathcal{T}^2$) based on any arbitrary language instruction (e.g., $l \sim \mathcal{T}$).
> Accordingly, the true reward function associated with $f_{\psi}$ should be defined as a single multi-task reward function, encompassing all $\mathcal{R}^\mathcal{T}$ for each task $\mathcal{T}$ and conditioned by the language $l$.
>
> I believe the _regret_ should be defined with this ground-truth, multi-task, and language-conditional reward function, but I'm afraid the definition for this reward function is not clear.
>
> I would like to hear your thoughts on this point. Could you also share some intuition on how this ground-truth multi-task reward $\mathcal{R}\_{f_{\psi}}(\sigma \mid l)$ is structured in terms of each task-specific reward $\mathcal{R}^\mathcal{T}(\sigma)$?

---

> > ### Author Response · Authors · 2024-12-04
> > **Further response (Part 2/2)**
> >
> > **Q4: "I believe the  _regret_  should be defined ... is not clear."**
> >
> > In our preference model, $R(v|l)$ is sufficient to determine whether $\sigma^1$ is preferred over $\sigma^2$ in a specific language description $l$. As a result, the regret should be defined by ground-truth and language-conditional reward function. And the multi-task property is implicitly contained in the language description.
> >
> > The regret is defined using the state-action value function ($Q$) and the state value function ($V$) under the optimal policy $\pi^*$. Specifically:
> > ${\rm regret}(\sigma \mid l) = \sum_{t \in \sigma} \big[ Q^*(s_t, a_t \mid l) - V^*(s_t \mid l) \big]$,
> > where $Q^*(s_t, a_t \mid l)$ is the optimal action-value (expected cumulative reward) for state-action pair $(s_t, a_t)$ given instruction $l$. $V^*(s_t \mid l)$ is the optimal state value (expected cumulative reward) for state $s_t$ under instruction $l$. The preference model $f_\psi(\sigma \mid l)$ is optimized to approximate the negative regret over the trajectory, i.e.:
> > $f_\psi(\sigma \mid l) \approx -{\rm regret}(\sigma \mid l) = -\sum_{t \in \sigma} \big[ Q^*(s_t, a_t \mid l) - V^*(s_t \mid l) \big]$.
> > The learned preference distribution is then defined as:
> > $P_\psi[\sigma_1 \succ \sigma_2 \mid l] = \frac{\exp(f_\psi(\sigma_1 \mid l))}{\exp(f_\psi(\sigma_1 \mid l)) + \exp(f_\psi(\sigma_2 \mid l))}$.
> > This distribution reflects the likelihood of one trajectory being preferred over another based on their relative regrets under the language-conditioned reward.
> >
> > **Q4: "Could you also share some intuition on ... task-specific reward?"**
> >
> > The structure of the single-step reward function $r(s, a \mid l)$ can be understood as a combination of task-specific information and language-conditioned alignment. Specifically:
> >
> > - Task-Specific Contribution: For a given task $\mathcal{T}$, the reward evaluates how well the state-action pair $(s, a)$ aligns with the task's objective. For instance, in a "drawer-closing" task, the reward may depend on metrics such as the drawer’s position at state $s$ and the action $a$ taken towards closing it.
> > - Language-Conditioned Contribution: The reward further incorporates alignment between the state-action features and the semantics of the instruction $l$. For example: If $l$ specifies "close the drawer," the reward increases for states and actions associated with drawer-closing. If $l$ specifies "open the window," states and actions related to window-opening will have higher rewards.
> >
> > Thus, the single-step reward $r(s, a \mid l)$ can be expressed as:
> > $r(s, a \mid l) = \alpha r^\mathcal{T}(s, a) + \beta \text{Align}(s, a, l)$, where $\alpha$ and $\beta$ balance the task-specific and language-conditioned contributions. $\text{Align}(s, a, l)$ quantifies the semantic match between the state-action pair and the language instruction.

---

> ### Author Response · Authors · 2024-12-04
> **Further response (Part 1/2)**
>
> We thank Reviewer LWSo for the insightful feedback. We provide our point-wise responses below.
>
> **Q2-continued**
>
> Thanks for your feedback. We have updated the medium-optimality videos across the 50 Meta-World tasks on our website. Please refer to the website for details.
>
> ---
>
> **W1-continued: "Can we say that $f_\psi$ can be considered as ... different preference dataset?"**
>
> Thanks for the question. The answer is yes if the preference labels are the same for two preference datasets. Let us define two preference datasets in the same format as $\{(v_1, v_2, l, y)\}$, where video pair ($v_1$, $v_2$)  and language description $l$ can be sampled randomly from their space, and $y$ denote whether $v_1$ is preferred than $v_2$ in such a language description. Then, (1) in the ground-truth preference dataset $D_r$, the preference label $y$ is given by comparing the ground-truth accumulated reward on videos, i.e., $R(v_1|l)$ and $R(v_2|l)$; while (2) in the preference dataset $D_{VLP}$ of VLP, the preference label $y$ is determined by the optimality (if $v_1$, $v_2$ are from the same task defined by $l$) or the video-language alignment (if $v_1$, $v_2$ are from different task and one video belongs to task defined by $l$, or both videos not belong to task $l$), which are defined by the pseudo-labels $\tilde{y}\in\\{y^{\text{ITP}}, y^{\text{ILP}}, y^{\text{IVP}}\\}$ in VLP. We remark that there are also other cases beyond these kinds of preferences defined in pseudo-labels of VLP, in which we use the generalization of learned preference model $f_\psi(v|l)$ to define its preference label.
>
> In an ideal case, if the preference label of $D_r$ and $D_{VLP}$ are the same, then $f_\psi$ can be considered as the same as $g_\varphi$. In practice, $D_r$ and $D_{VLP}$ can be different in some cases beyond the definition of pseudo-labels $\tilde{y}$ depending on the generalization ability of the learned preference model.
>
> **W1-continued: "But this result, suggesting that the parameterized immediate reward ... sounds weird."**
>
> Thanks for the problem. We believe the definition of "immediate reward function" as $h(\sigma):= r(s_1^{\sigma}, a_1^{\sigma})$ is wrong since the immediate reward function can only be defined in a single time step. Specifically, the correct definition can be (1) $h(s_1, a_1):= r(s_1^{\sigma}, a_1^{\sigma})$ as the single-step reward function and $h(\sigma):=\sum_\sigma r(s_t^{\sigma}, a_t^{\sigma})$, which is the ordinary BT model. (2) $h(\sigma):=\sum_\sigma Q^*(s_t^{\sigma}, a_t^{\sigma})$ or $h(\sigma):=\sum_\sigma Q^*(s_t^{\sigma}, a_t^{\sigma})-V^*(s_t^{\sigma})$, where the two types of $h(\sigma)$ are actually the same in regret-based preference model since $V^*(s_t^{\sigma})$ is a constant. In the second case, $h(\sigma)$ approximates the negative regret, as we highlight in our theoretical analysis.
>
> ---
>
> **Q4: "Additionally, it would be helpful to ... learned preference distribution"**
>
> Thanks for the comment. Indeed, $f_\psi$ is defined as a multi-task preference model that can compare two segments from either the same task or different tasks (e.g., $\sigma^1 \sim \mathcal{T}^1, \sigma^2 \sim \mathcal{T}^2$) based on any arbitrary language instruction (e.g., $l \sim \mathcal{T}$). As a result, the reward function $r(s, a \mid l)$ is defined as a single-step reward that evaluates the immediate quality of a state-action pair $(s, a)$ based on the task's goal and the provided language instruction $l$, and $R(v|l)=\sum_v r(s,a|l)$ is defined as the undiscounted return. For a given task $\mathcal{T}$, the reward function depends on how well the state-action pair aligns with the task objective under the guidance of the language instruction. For example: If the instruction $l$ specifies "close the drawer," the reward for states where the drawer is closer to being closed will be higher. If the instruction $l$ specifies "open the window," the reward will prioritize states corresponding to window opening.
> The reward is inherently language-conditioned, which allows it to generalize across tasks. Formally, for a specific task $\mathcal{T}$, the single-step reward function $r(s, a \mid l)$ aligns with the task-specific reward $r^\mathcal{T}(s, a)$, but conditioned on the instruction $l$. For multi-task learning, the reward function is unified as: $r(s, a \mid l) = r^\mathcal{T}(s, a)$, where $\mathcal{T}$ is implicitly determined by the language instruction $l$.

---

### Official Review · Reviewer_ARLJ · 2024-11-04

**Soundness:** 3
**Presentation:** 3
**Contribution:** 3
**Rating:** 6
**Confidence:** 4

**Summary:**

This work we propose a vision-language preference (VLP) learning that uses a vision-language model to provide preference feedback. It defines three types of language-conditioned preferences and contributes a vision-language preference dataset. The framework is evaluated on the Meta-World Benchmark.

**Strengths:**

1. This work proposes three forms of language-conditioned preferences: ITP, ILP and IVP.

2. This work proposes a framework vision-language preference learning with theoretic analysis of its behavior.

3. Experiments are well organized to answer four key questions.

4. Experiments show that the proposed VLP leads to better performance than other state-of-the-art baselines.

**Weaknesses:**

1. In the experiments, only a single benchmark, Meta-World, is used.
- This is limited to show the generality of the proposed preference learning framework.
- Related works in section 4 have tested on several different environments.

2. Only five tasks in the Meta-work are evaluated among 50 tasks.
- This set of test tasks is not so challenging compared to 45 training tasks.

3. Why are RL-VLM-F and CriticGPT not compared?

4. The dataset construction itself may not be a notable contribution.
- The trajectory sampling pipeline is rather simple, so its diversity may be unclear.
- The dataset size is not big.

5. Figures can be improved.
- Fig. 2 is somewhat standard and conveys only limited value for the novelty.
- In Fig.3, the attentions are not clearly seen.

**Questions:**

Please refer to Weaknesses.

---

> ### Author Response · Authors · 2024-11-23
> **Rebuttal**
>
> We thank reviewer ARLJ for the positive support and insightful comments. We will give our point-wise responses below.
>
> **W1: "In the experiments, ... on several different environments."**
>
> > **A**: Please refer to the subsequent glogal response.
>
> **W2: "Only five tasks in the Meta-work are evaluated among 50 tasks. This set of test tasks is not so challenging compared to 45 training tasks."**
>
> > **A**: Thank you for your valuable feedback. The test tasks used in our paper are motivated by the test tasks in prior preference-based RL works [1,2,3,4,5]. To address this, we conduct experiments on the ML45 benchmark, training the vision-language preference model on its training tasks and evaluating on its test tasks. The results shown below demonstrate the strong generalization capability of our method on unseen tasks in ML45. This reinforces the robustness and adaptability of our framework regardless of task split. We have updated the manuscript and included these results and discussions in Appendix E.
> >
> > | Task | VLP Accuracy |
> > | - | - |
> > | Bin Picking | 95.0 |
> > | Box Close | 90.0 |
> > | Door Lock | 100.0 |
> > | Door Unlock | 100.0 |
> > | Hand Insert | 100.0 |
> > | Average | 97.0 |
>
> **W3: "Why are RL-VLM-F and CriticGPT not compared?"**
>
> > **A**: RL-VLM-F and CriticGPT are both online preference-based RL algorithms, while our method focues on offline preference-based RL setting.
>
> **W4: "The dataset construction itself may ... size is not big."**
>
> > **A**: Thank you for raising this concern. While the trajectory sampling pipeline is straightforward, we would like to clarify that the contribution lies in addressing a critical gap in the field: the lack of high-quality, multi-modal preference datasets for vision-language preference learning. It introduces implicit preference labels conditioned on diverse language instructions, which are currently unavailable in existing benchmarks. By including trajectories of three distinct optimality levels (expert, medium, random) and pairing them with multiple language instructions, we ensure a varied and rich dataset. Our goal was to demonstrate the effectiveness of language-conditioned preference learning, not merely to build a large dataset. Future work could expand the dataset scale and diversity to facilitate broader research.
>
> **W5: "Figures can be improved."**
>
> > **A**: Thank you for this feedback. Regarding Fig. 2, we understand it may appear standard. However, its primary purpose was to clearly present the architecture vision-language preference model.
> > For Fig. 3, we have improved the clarity of the attention maps to illustrate the cross-modal attention mechanism. Please refer to the revised manuscript for these updates.
>
> **References**
>
> [1] PEBBLE: Feedback-Efficient Interactive Reinforcement Learning via Relabeling Experience and Unsupervised Pre-training. ICML 2021.
>
> [2] SURF: Semi-supervised Reward Learning with Data Augmentation for Feedback-efficient Preference-based Reinforcement Learning. ICLR 2022.
>
> [3] Reward Uncertainty for Exploration in Preference-based Reinforcement Learning. ICLR 2022.
>
> [4] Few-Shot Preference Learning for Human-in-the-Loop RL. CoRL 2023.
>
> [5] PEARL: Zero-shot Cross-task Preference Alignment and Robust Reward Learning for Robotic Manipulation. ICML 2024.

---

> ### Author Response · Authors · 2024-12-03
> **A gentle reminder**
>
> Dear Reviewer ARLJ,
>
> As the deadline of the discussion period draws near, we would greatly appreciate your attention to our rebuttal. We would like to know whether we have adequately addressed your concerns. Your feedback is crucial to us, and we value the opportunity to address any concerns you have raised. The answer to **W1** has been provided in the global response. If you have any further questions, we are more than happy to discuss.
>
> Thank you for your time and consideration.
>
> Best regards,
>
> Authors

---

### Author Response · Authors · 2024-12-03
**Global response**

We are sincerely grateful to all reviewers for their careful evaluation and constructive feedback on our paper. We appreciate reviewers recognizing the contributions of our work, including the novelty of the vision-language preference [ARLJ, LWSo, R4Zt], the writing of the paper [ARLJ, NSgJ, R4Zt], and the the strong empirical results [ARLJ, LWSo, NSgJ].

In response to reviewer comments, we have added new experiments and results to strengthen the paper in the PDF, and all modifications are highlighted in **orange**. We provide global response for the following questions:

**More benchmark results [ARLJ, NSgJ]**

> We acknowledge the importance of evaluating our method on more complex task domains and conduct further experiments on ManiSkill2 [1]. We leverage *MoveBucket-v1*, *OpenCabinetDrawer-v1*, *PegInsertionSide-v0*, *PickCube-v0*, *PickSingleEGAD-v0*, *PlugCharger-v0*, *StackCube-v0*, and *TurnFaucet-v0* as training tasks and evaluate the  of VLP on *LiftCube-v0*, *OpenCabinetDoor-v1*, *PushChair-v1* tasks. The following table summarize the average VLP label accuracy on the three test tasks compared to scripted labels and the results demonstrate the strong generalization capabilities of VLP. To provide a comprehensive evaluation of VLP on ManiSkill2 benchmark, we will conduct further experiments and add the results in the future version.
>
> | Task | VLP Accuracy |
> | - | - |
> | LiftCube-v0 | 100.0 |
> | OpenCabinetDoor-v1 | 100.0 |
> | PushChair-v1 | 93.8 |
> | Average | 97.9 |

**Train-test task split [R4Zt]**

> 1) We conduct experiments on the standard ML45 benchmark, training the vision-language preference model on its training tasks and evaluating on its test tasks. The results shown below demonstrate the strong generalization capability of our method on unseen tasks in ML45. This reinforces the robustness and adaptability of our framework regardless of task split. We have updated the manuscript and included these results and discussions in Appendix E.
>
> | Task | VLP Accuracy |
> | - | - |
> | Bin Picking | 95.0 |
> | Box Close | 90.0 |
> | Door Lock | 100.0 |
> | Door Unlock | 100.0 |
> | Hand Insert | 100.0 |
> | Average | 97.0 |
>
> 2) We believe that there is no universally accepted "gold standard" for how to split tasks in multi-task setting and the train-test split is just a design choice instead of a standardized procedure.  CriticGPT [2], which is prior preference-based RL method evaluated in similar multi-task setting, also uses the Meta-World benchmark and manually selects its own train-test task split. Lift3D [3] also select 15 custom tasks for experimental evaluation. Additionally, the baselines are evaluated using the same split for fair comparison.

We believe these new results and clarification will help address the concerns about the evaluation benchmark and train-test task split. We thank the reviewers' time and feedback in improving the quality of our work and we hope the revisions further highlight the contributions made. Please let us know if any clarification or additional experiments would further strengthen the paper. We would be happy to incorporate all these suggestions in the final version.

**References**

[1] Maniskill2: A unified benchmark for generalizable manipulation skills.

[2] Enhancing Robotic Manipulation with AI Feedback from Multimodal Large Language Models.

[3] Lift3D Foundation Policy: Lifting 2D Large-Scale Pretrained Models for Robust 3D Robotic Manipulation.

---

### Meta-Review · Area_Chair_P4Bs · 2024-12-25

**Metareview:**

The paper introduces a video-based, vision-language-interleaved preference learning approach for robotic control. Reviewers raised concerns, particularly regarding the limited novelty in the architecture and the weakness of the experimental results. While the paper shows promise, it is not recommended for acceptance in its current form. The authors are encouraged to address the reviewers' feedback and refine the work for submission to other venues.

**Additional Comments On Reviewer Discussion:**

The primary concerns revolve around the limited novelty of the model architectures and experiments.

---

### Decision · Program_Chairs · 2025-01-22

Reject